# Plasma tau biomarkers for biological staging of Alzheimer's disease

Laia Montoliu-Gaya [1,16] ✉, Gemma Salvadó [2,16], Joseph Therriault[3,4], Johanna Nilsson [1], Shorena Janelidze [2], Sophia Weiner [1], Nicholas J. Ashton[1,5,6], Andrea L. Benedet[1], Nesrine Rahmouni[3,4], Juan Lantero-Rodriguez [1], Niklas Mattsson-Carlgren [2,7,8], Sebastian Palmqvist [2], Gunnar Brinkmalm [1], Erik Stomrud[2,8], Henrik Zetterberg [1,9,10,11,12,13], Johan Gobom [1], Pedro Rosa-Neto [3,4], Kaj Blennow [1,9,14,15,17] & Oskar Hansson [2,8,17] ✉

A blood biomarker-based staging system for Alzheimer's disease (AD) could improve the diagnosis, prognosis and identification of individuals most likely to benefit from specific therapies. Here, using targeted mass spectrometry, we measured six phosphorylated and six nonphosphorylated tau peptides in plasma from two independent cohorts: BioFINDER-2 and TRIAD ($n = 689$). We also analyzed the ratios of phosphorylated to nonphosphorylated peptides. Our results revealed that specific tau species became abnormal at different points along the disease continuum. Based on these findings, we developed a data-driven, blood-based staging model that demonstrated strong consistency across cohorts (>85% agreement in ≥90% initializations) and reflected changes in other AD biomarkers. These plasma-based stages were associated with clinical diagnoses, positron emission tomography-based stages and distinct patterns of longitudinal disease progression, including Aβ- and tau-positron emission tomography uptake, atrophy and cognitive decline. This study highlights the potential of tau blood-based biomarkers for biological staging in AD, offering a scalable tool for tracking disease progression and guiding clinical decisions.

Alzheimer's disease (AD) is the leading cause of dementia worldwide, affecting more than 55 million patients[1]. The recent approval of amyloid-β (Aβ)-targeting immunotherapies for the treatment of mild-to-moderate AD underscores the need for accessible biomarkers, not only for detecting the disease pathologies but also staging such pathologies for optimal treatment initiation and evaluating outcomes[2]. Staging systems that model disease progression can provide a framework for detecting and monitoring pathological changes over its long course and assist in clinical decision-making[3]. For example, evidence indicates that patients with AD who exhibit lower tau tracer uptake on positron emission tomography (PET) scans respond better to Aβ-targeting monoclonal antibody therapies, suggesting that this group may have the most favorable risk–benefit ratio for this

treatment[4,5]. It is likely that other therapies, such as tau-targeting therapies, will also prove to be more beneficial at certain biological disease stages. However, PET scans are costly and require specialized personnel and facilities[6,7]. Implementing a panel of fluid biomarkers, especially in blood, would substantially enhance the accessibility and cost-effectiveness of biomarker-based AD staging.

Although fluid biomarkers do not provide brain topographical information on Aβ and tau pathology, as imaging biomarkers do[8], emerging evidence suggests that various tau species could be used for staging the disease. In the AD brain, pathological hyperphosphorylation of tau occurs sequentially at different sites along the sequence of the protein[9,10]. As the disease advances, both the number of phosphorylation sites and the extent of their occupancy increase, aligning

with the gradual rise in the molecular weight of tau aggregates and overall disease progression[9]. This differential pattern of tau phosphorylation has been observed in cerebrospinal fluid (CSF), where site-specific phosphorylation changes follow distinct trajectories over time[11,12]. For instance, phosphorylations such as p-tau181, p-tau217 and p-tau231 begin to rise in CSF concurrently with the early increase in Aβ aggregates[11,13], while tau phosphorylation at position 205 may rise later and be more strongly associated with tau-PET measures[14,15]. Nonphosphorylated tau species, such as N-terminal tau fragments (NTA)[16,17], total tau (T-tau) measured with mid-domain sandwich assays[18], MTBR-tau243 (refs. 19,20) or nonphosphorylated tau peptides[11,21], have been shown to increase in CSF at later stages, with even stronger associations with tau-PET and cognition. The distinct emergence of phosphorylated and nonphosphorylated tau biomarkers has been leveraged to create a CSF staging model[22], where more advanced stages correlate with higher amyloid and tau-PET uptake and greater cognitive impairment. These CSF stages also predict the longitudinal trajectories of imaging biomarkers and clinical progression. Yet, the potential use of tau blood-based biomarkers for staging AD remains in its early stages. Multiple studies using different assays and platforms suggest that the pattern observed in CSF might also be detectable in blood[23-26]. The Alzheimer's Association (AA) recently published revised criteria for the diagnosis of AD[27,28], updating the previous guidelines published in 2018[29]. These new criteria also include a biomarker framework for staging AD using amyloid and tau-PET imaging biomarkers, as well as a conceptual biological staging model based on fluid biomarkers. However, further investigation and validation are necessary to confirm the utility of fluid tau biomarkers, particularly those in blood, for staging AD—an advancement that would be highly valuable for patient management.

In this study, we examined how various plasma tau biomarkers become abnormal across the AD continuum and evaluated their ability to stage the disease. We utilized a targeted mass spectrometry (MS) method that systematically quantifies multiple phosphorylated and nonphosphorylated plasma tau species in a single analysis[30,31]. Specifically, we measured levels of six phosphorylated and six nonphosphorylated tau peptides in plasma samples from two independent cohorts of sporadic AD: the Swedish BioFINDER-2 study (n = 549) and Translational Biomarkers in Aging and Dementia (TRIAD) cohort (n = 140). We first assessed the point at which the levels of different plasma tau biomarkers became abnormal over the course of the clinical disease stages in the BioFINDER-2 cohort and developed a staging model based on selected biomarkers. This model was subsequently applied to the BioFINDER-2 cohort and independently validated in the TRIAD cohort. We then compared the distribution of imaging biomarkers and cognitive measures across the plasma stages, and explored associations between plasma stages, clinical diagnosis and imaging stages in accordance with the new AA criteria[27,28]. Finally, using longitudinal data in the BioFINDER-2 cohort, we determined the rate and trajectory of Aβ and tau-PET accumulation, cognitive decline and neurodegeneration for individuals at each plasma stage.

## Results

### Participant characteristics and plasma tau biomarker measures

The BioFINDER-2 cohort included 549 individuals: 191 were cognitively unimpaired Aβ negative (CU−), 82 were cognitively unimpaired Aβ positive (CU+), 76 were mildly cognitively impaired and Aβ positive (MCI+) and 80 had AD dementia and were Aβ positive (ADdem+). A subset of participants exhibited other neurodegenerative diseases than AD (95 Aβ negative and 18 Aβ positive). The average age was 70.4 ± 12.5 years (mean ± s.d.), 284 (51.7%) were women and 281 (51.3%) were *APOE* ε4 carriers. The TRIAD cohort included a total of 140 participants, distributed in groups across the AD continuum: 34 CU−, 25 CU+, 35 MCI+ and 28 AD dementia, while 7 exhibited other neurodegenerative diseases. The average age was 71.5 ± 6.5 years, 58 (41.4%) were women and 61 (43.6%) were *APOE* ε4 carriers. Demographic information is presented in Table 1.

**Table 1 | Participant characteristics in the BioFINDER-2 and TRIAD cohorts**

| | BioFINDER-2 (*n*=549) | TRIAD (*n*=140) | *P* value |
|---|---|---|---|
| Age, years | 70.4 (12.5) | 71.5 (6.5) | 0.140 |
| Women, *n* (%) | 284 (51.7%) | 58 (41.4%) | 0.176 |
| Education years | 12.4 (3.7) [*n*=545] | 15.6 (3.49) | <0.001 |
| *APOE* ε4 carriership, *n* (%) | 281 (51.3%) [*n*=548] | 61 (43.6%) | 0.125 |
| Diagnosis, *n* (%) | | | |
| CU− | 191 (34.8%) | 34 (24.3%) | |
| CU+ | 82 (14.9%) | 25 (17.9%) | |
| MCI+ | 76 (13.8%) | 35 (25.0%) | <0.001 |
| AD dementia | 80 (14.6%) | 28 (20.0%) | |
| Cognitively impaired not due to AD | 113 (20.6%) | 7 (5.0%) | |
| Other | 7 (1.3%) | 11 (7.9%) | |
| Aβ positivity, *n* (%) | 256 (47.2%) [*n*=542] | 88 (62.9%) [*n*=129] | <0.001 |
| Tau-PET positivity, *n* (%) | 150 (27.5%) [*n*=545] | 54 (38.6%) [*n*=127] | 0.001 |

Mean (s.d.) shown unless otherwise specified. In characteristics in which some data were missing, we include the final data sample in square brackets.

In BioFINDER-2, a subset of participants had available longitudinal data on Aβ-PET (n = 368), tau-PET (n = 480) and cortical thickness (n = 457).

Plasma tau biomarkers were measured in the BioFINDER-2 and TRIAD cohorts using a targeted MS method[30]. This approach quantifies the levels of six phosphorylated tau species (p-tau181, p-tau199, p-tau202, p-tau205, p-tau217 and p-tau231) and six nonphosphorylated tau peptides (tau195-209, tau212-221, 0N-tau, 1N-tau, PNS131-138 and PNS275-291). The nonphosphorylated tau species 195-209 and 212-221 are common to all tau isoforms, whereas the 0N and 1N-tau peptides are specific to the 0N and 1N tau isoforms, respectively. The PNS peptides are unique to the PNS-tau isoform, which is primarily expressed in the peripheral nervous system (PNS)[32].

### Plasma tau biomarkers emerge differentially across the AD continuum

We first examined how the levels of plasma tau biomarkers varied across diagnostic groups based on cognitive status and Aβ status (Fig. 1, Extended Data Fig. 1 and Supplementary Table 1). Apart from the quantified peptides, we additionally included in the analysis the ratios between the phosphorylated peptides p-tau217 and p-tau205 and their respective nonphosphorylated forms: p-tau217/212-221 (p-tau217r) and p-tau205/195-209 (p-tau205r). Using a threshold based on the mean plus 1.96 standard deviations of the CU− group (which corresponds to its 95% confidence interval (CI)) to define biomarker abnormality, we found that the different tau species showed significant changes to positivity at distinct points across the continuum. Plasma p-tau217 (*P* < 0.001), p-tau217r (*P* < 0.001) and p-tau231 (*P* < 0.001) exhibited abnormal levels in the CU+ stage. Plasma p-tau205 (*P* < 0.001) and p-tau205r (*P* < 0.001) values became abnormal at the MCI+ stage, while p-tau181 (*P* < 0.001) and 0N-tau (*P* = 0.001) levels changed to positivity in the ADdem+ group. Other p-tau biomarkers, including p-tau199, p-tau202 and the nonphosphorylated species tau195-209, tau212-221 and 1N-tau, presented some significant increases among diagnostic groups, but never above the positivity threshold. The peripheral-specific (PNS131-138 and PNS275-291) tau peptides did not show any significant change in their levels with disease progression.

### Plasma tau biomarker staging model

Based on these findings, we developed a plasma staging model using biomarkers with distinct abnormal changes along the continuum. Among biomarkers with changes to abnormality occurring at the same

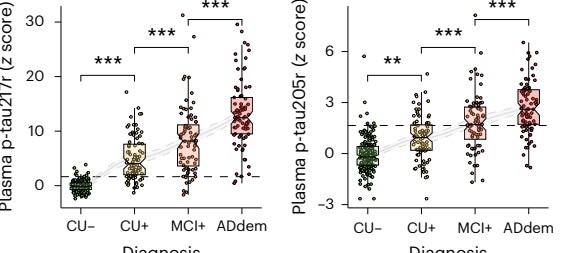

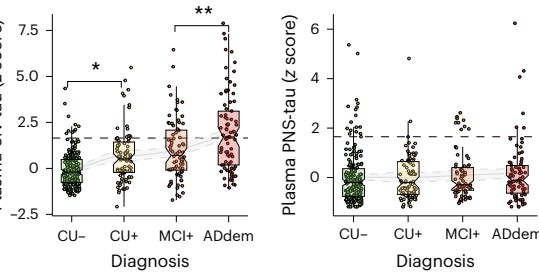

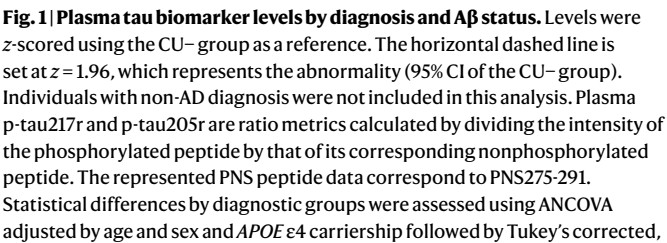

**Fig. 1 | Plasma tau biomarker levels by diagnosis and Aβ status.** Levels were z-scored using the CU− group as a reference. The horizontal dashed line is set at $z = 1.96$, which represents the abnormality (95% CI of the CU− group). Individuals with non-AD diagnosis were not included in this analysis. Plasma p-tau217r and p-tau205r are ratio metrics calculated by dividing the intensity of the phosphorylated peptide by that of its corresponding nonphosphorylated peptide. The represented PNS peptide data correspond to PNS275-291. Statistical differences by diagnostic groups were assessed using ANCOVA adjusted by age and sex and *APOE* ε4 carriership followed by Tukey's corrected, post-hoc pairwise comparisons (two-sided analysis). In the figure, only differences between contiguous groups are shown. Boxplots summarize the data distribution, showing the median (central line), interquartile range (IQR; box) and whiskers extending to 1.5× the IQR. All individuals with available data were included ($n = 549$). Exact $P$ values can be found in Supplementary Table 1. *$P < 0.05$, **$P < 0.005$, ***$P < 0.001$. CU−, cognitively unimpaired Aβ-negative; CU +, cognitively unimpaired Aβ-positive; FDR, false discovery rate; MCI +, mild cognitive impairment Aβ-positive; ADdem, Alzheimer's disease dementia.

clinical stage, p-tau217r was chosen over p-tau231 and 0N-tau over p-tau181, due to their greater significant changes across the disease continuum (Supplementary Table 1). In addition, the ratios p-tau217r and p-tau205r were selected over the phosphorylated tau species alone because ratios of p-tau to nonphosphorylated tau may help minimize confounding effects from medical comorbidities such as kidney disease. To explore this, we performed a sensitivity analysis stratifying participants by chronic kidney dysfunction status based on estimated glomerular filtration rate measures (Supplementary Table 2). We found that all tau biomarkers, including PNS-tau, were significantly altered in individuals with kidney dysfunction, but this effect was mitigated when using the phospho/nonphospho ratios, corroborating previous findings[33].

We first normalized the selected plasma biomarkers (p-tau217r, p-tau205r and 0N-tau), dividing by the median of the CU− group, in the two cohorts independently. We then performed *k*-means clustering to classify each individual into one of four blood-based stages, using these three biomarkers (Methods). This can be visualized as four distinct clusters within a three-dimensional space, where each biomarker represents one of the axes (Extended Data Fig. 2). The final model included a negative stage (that is, stage 0), which corresponds to normality in all biomarkers, and three positive stages (that is, stages 1–3). Locally estimated scatterplot smoothing (LOESS) representation of the change of the plasma biomarkers at each stage showed that p-tau217r p-tau205r and 0N-tau became abnormal consecutively and in accordance with the plasma stages (Extended Data Fig. 3) and presented differential correlations among tau markers at each plasma stage (Extended Data Fig. 4).

The blood biomarker stages were not associated with years of education (BioFINDER-2 $P = 0.083$, TRIAD $P = 0.210$), but an increase in the proportion of *APOE* ε4 carriers was observed with advancing stages (BioFINDER-2 and TRIAD $P < 0.001$). A slightly increasing trend in the percentage of women with plasma stages was observed in BioFINDER-2 ($P = 0.047$), which was not significant in TRIAD ($P = 0.063$) (Extended Data Fig. 5).

To assess the replicability of the model, we reclassified each individual with different starting points ($n = 5,000$) and determined the percentage of occasions each individual was placed in the plasma-based same stage (Supplementary Table 3). Notably, >85% of individuals were classified in the same stage ≥90% of the occasions. In addition, we tested the model using biomarker concentrations instead of MS-calculated ratios, as they may be more applicable in clinical settings. Both models showed a relatively high correspondence (BioFINDER-2: 0.73, TRIAD: 0.69; Supplementary Table 4).

## Plasma biomarker stages show a consistent pattern in relation to imaging biomarkers and cognition

We then evaluated the relationship between the plasma stages in our model and key AD markers, including insoluble Aβ aggregates (Aβ-PET), insoluble tau aggregates (tau-PET) at mediotemporal (MTL) and temporal neocortex (NeoT) regions, neurodegeneration (cortical thickness) and cognitive performance. Regarding the plasma stages, Aβ-PET showed the earliest changes toward abnormality, followed by tau-PET MTL, tau-PET NeoT, cognitive scores and finally, cortical thickness (Extended Data Fig. 6). In addition, the model was proven accurate to predict Aβ-PET (AUC (95% CI) = 84 (81 to 87)) and tau-PET (AUC (95% CI) = 90 (87 to 93)) status as well as AD-related cognitive symptoms (AUC (95% CI) = 88 (84 to 92)) (Extended Data Fig. 7).

A consistent pattern of changes in AD pathology biomarkers was observed across stages in both the BioFINDER-2 and TRIAD cohorts, supporting the validity of the model (see representation in Fig. 2, quantitative cross-stage profiles in Supplementary Table 5 and analysis of group differences in Supplementary Table 6). In the BioFINDER-2 cohort, Aβ-PET levels exhibited the first significant increase between plasma stages 0 and 1 ($\beta = 0.37$ (0.14 to 0.59), $P < 0.001$), followed by further elevation at stage 2 ($\beta = 1.34$ (1.03 to 1.65), $P < 0.001$), after which they plateaued at stage 3 ($\beta = -0.04$ (−0.59 to 0.5), $P = 0.99$). MTL tau-PET showed significant changes from stage 1 to stage 2 ($\beta = 1.30$ (1.04 to 1.56), $P < 0.001$). NeoT tau-PET exhibited significant differences between stages 1 and 2 ($\beta = 1.13$ (0.87 to 1.4), $P < 0.001$) and further increases between stages 2 and 3 ($\beta = 0.99$ (0.63 to 1.35), $P < 0.001$). Cortical thickness progressively decreased with advancing plasma stages, showing significant reductions between stages 0 and 1 ($\beta = -0.27$ (−0.52 to −0.02), $P = 0.027$) and stages 1 and 2 ($\beta = -0.84$ (−1.15 to −0.53), $P < 0.001$). Cognitive performance exhibited significantly lower scores between stages 1 and 2, and stages 2 and 3 for the modified preclinical Alzheimer's cognitive composite (mPACC; $P < 0.008$) and Mini-Mental State Examination (MMSE; $P < 0.001$).

Similarly, in the TRIAD cohort, Aβ-PET levels showed marked increases between stages 0 and 1 ($\beta = 0.63$ (0.25 to 1), $P < 0.001$) and stages 1 and 2 ($\beta = 1.11$ (0.72 to 1.51), $P < 0.001$), before leveling off ($\beta = 0.05$ (−0.37 to 0.47), $P = 0.99$). MTL tau-PET displayed significant changes between stages 1 and 2 ($\beta = 0.71$ (0.25 to 1.18), $P = 0.001$), NeoT tau-PET exhibited significant differences between stages 1 and 2 ($\beta = 0.73$ (0.24 to 1.23), $P = 0.001$), with further elevations at stage 3 ($\beta = 0.69$ (0.15 to 1.22), $P = 0.006$). Cortical thickness progressively decreased with advancing plasma stages, with significant changes observed between stages 2 and 3 ($\beta = -0.66$ (−1.24 to −0.07), $P = 0.022$). Cognitive performance showed a trend to higher scores for the Clinical

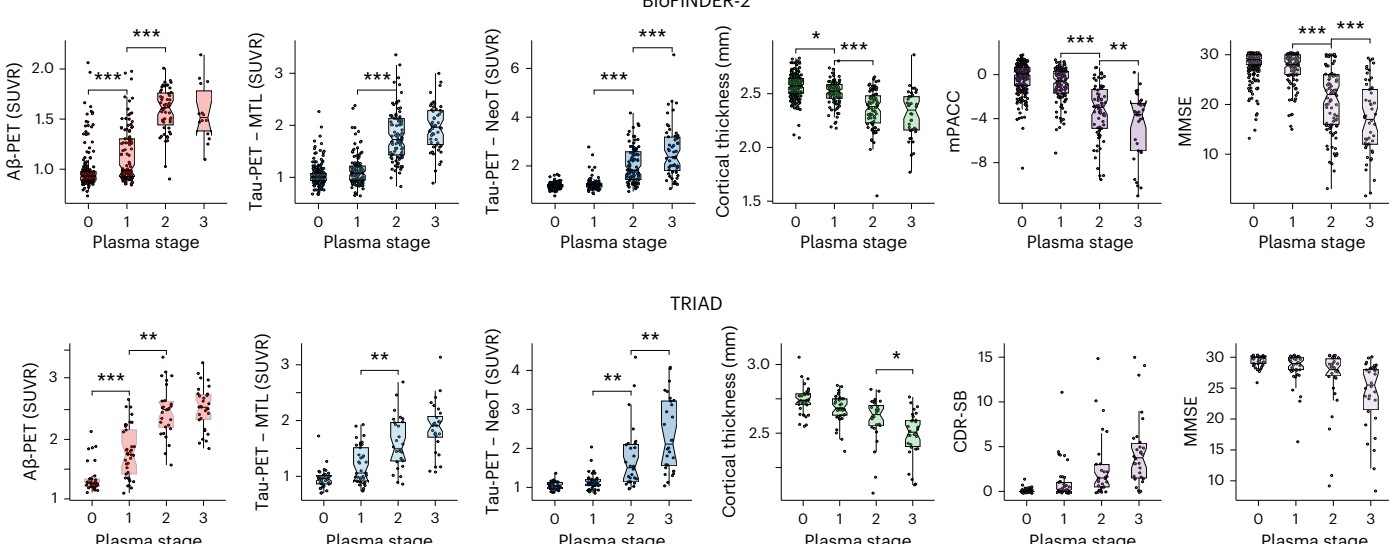

**Fig. 2 | AD biomarkers by plasma stage.** Biomarkers of Aβ, early (MTL) and intermediate (NeoT) tau pathology measured by PET, neurodegeneration (cortical thickness) and sensitive (mPACC or CDR-SB) and global cognition (MMSE) by stages defined by plasma biomarkers. Plasma stages were created in the main (BioFINDER-2) cohort and validated in the replication cohort (TRIAD). All individuals with available data were included in the Aβ- and tau-PET analyses, but individuals with non-AD diagnosis were excluded from cortical thickness and cognitive tests to avoid bias. Statistical differences by plasma stage were assessed using ANCOVA adjusted by age and sex and *APOE* ε4 carriership followed by Tukey's corrected, post-hoc pairwise comparisons (two-sided analysis). In the figure, only differences between contiguous groups are shown. Boxplots summarize data distribution, showing the median (central line), interquartile range (IQR; box) and whiskers extending to 1.5× the IQR. All individuals with available data were included (Aβ-PET: $n = 370$ in BioFINDER-2 and $n = 129$ in TRIAD, MTL tau-PET and NeoT tau-PET: $n = 476$ in BioFINDER-2 and $n = 127$ in TRIAD, cortical thickness: $n = 450$ in BioFINDER-2 and $n = 129$ in TRIAD, mPACC/CDR-SB: $n = 476$ in BioFINDER-2 and $n = 135$ in TRIAD, MMSE: $n = 529$ in BioFINDER-2 and $n = 138$ in TRIAD). Exact $P$ values can be found in Supplementary Table 6. *$P < 0.05$, **$P < 0.005$, ***$P < 0.001$.

Dementia Rating (CDR) sum of boxes (CDR-SB) and lower for MMSE across plasma stages.

### Plasma tau stages are associated with clinical diagnosis and PET-based stages

Next, we examined how the plasma tau stages were associated with clinical diagnostic groups within the AD continuum. In the BioFINDER-2 cohort ($\chi^2 = 297$, d.f. 9, $P < 0.001$; Fig. 3a), plasma stage 0 predominantly (90%) comprised cognitively unimpaired individuals (Aβ− or Aβ+). In stage 1, this percentage decreased to 80%, with an increase in Aβ+ participants to 56%. Stage 2 primarily consisted of Aβ+ individuals (99%), with 78% showing cognitive impairment (36% MCI+ and 42% ADdem+). By stage 3, 100% of participants were Aβ+, with 92% exhibiting cognitive impairment, including a larger proportion of the dementia group (25% MCI+ and 67% ADdem+).

We then assessed the association between the plasma tau stages and Aβ- and tau-PET imaging stages based on the new AA criteria[28] ($\chi^2 = 312.83$, d.f. 9, $P < 0.001$; Fig. 3b). In these criteria, individuals are categorized on the basis of their Aβ-PET status (negative or positive) and tau-PET uptake in in MTL and NeoT regions as follows: negative for both amyloid and tau-PET (A−T$_2$−), initial (Aβ-PET positive and tau-PET negative, A+T$_2$−), early (Aβ-PET positive and tau-PET MTL positive, A+T$_{2MTL}$+), intermediate (Aβ-PET positive and tau-PET MTL and NeoT positive, A+T$_{2MOD}$+) and advanced (Aβ-PET positive and tau-PET MTL and NeoT positive with high uptake in NeoT, A+T$_{2MOD}$+A+T$_{2HIGH}$+). In plasma stage 0, 78% of individuals were Aβ- and tau-PET negative. In stage 1, the negative Aβ- and tau-PET group was reduced to 43%, 37% were Aβ positive and tau-PET negative, and 21% were Aβ- and tau-PET positive. At stage 2, 90% of individuals were tau-PET positive, with 32% classified as early or intermediate and 58% as advanced. In stage 3, 94% of participants were tau-PET positive, with 74% in the advanced category.

These results were replicated in the TRIAD cohort for clinical diagnosis ($\chi^2 = 109$, d.f. 9, $P < 0.001$; Fig. 3c) and for PET AA criteria ($\chi^2 = 103$,

d.f. 9, $P < 0.001$; Fig. 3d). In plasma stage 0, most participants (94%) were cognitively unimpaired. This percentage decreased to 61% in stage 1. In stages 2 and 3, 100% of participants were Aβ+, with 79% and 90% being cognitively impaired, respectively. In terms of imaging stages, 81% of individuals in plasma stage 0 were both Aβ- and tau-PET negative. By stage 1, 52% of participants were tau-PET positive. In stage 2, 86% of individuals were tau-PET positive, and by stage 3, tau-PET positivity reached 96%, with 67% classified as having advanced tau-PET uptake.

These analyses were also performed including participants outside the AD continuum and are shown in Extended Data Fig. 8.

### Individuals in different plasma tau stages differ in longitudinal trajectories

Longitudinal data from the BioFINDER-2 cohort (Supplementary Table 7) were used to examine whether plasma tau stages were associated with longitudinal Aβ-PET and tau-PET uptake—in both MTL and NeoT regions (Fig. 4)—as well as cortical thickness and cognitive decline (Fig. 5).

The evaluation of rate of change in Aβ-PET over time showed that participants in plasma stage 0 accumulated Aβ-PET faster than those in stage 1 ($\beta_{0vs1} = 0.06$ (0.04 to 0.09), $P < 0.001$). Longitudinal tau-PET of participants in stages 0 and 1 did not reveal any change over time—neither in MTL ($\beta_{0vs1} = 0.02$ (0.00 to 0.05), $P = 0.062$) nor in NeoT regions ($\beta_{0vs1} = 0.02$ (−0.01 to 0.05), $P = 0.122$). By contrast, those in stage 2 increased in signal tau-PET significantly faster than those in stage 1 in both regions of interest (MTL: $\beta_{1vs2} = 0.11$ (0.08 to 0.15), $P < 0.001$; and NeoT: $\beta_{1vs2} = 0.23$ (0.19 to 0.27), $P < 0.001$), and individuals in stage 3 exhibited an even faster increase in tau-PET signal than those in stage 2 (MTL: $\beta_{2vs3} = 0.06$ (0.01 to 0.11), $P = 0.031$; and NeoT: $\beta_{2vs3} = 0.13$ (0.08 to 0.19), $P < 0.001$).

Longitudinal reductions in cortical thickness were observed at every plasma stage, with the rate of decline differing significantly between consecutive stages ($\beta_{1vs0} = −0.05$ (−0.09 to −0.01), $P = 0.020$;

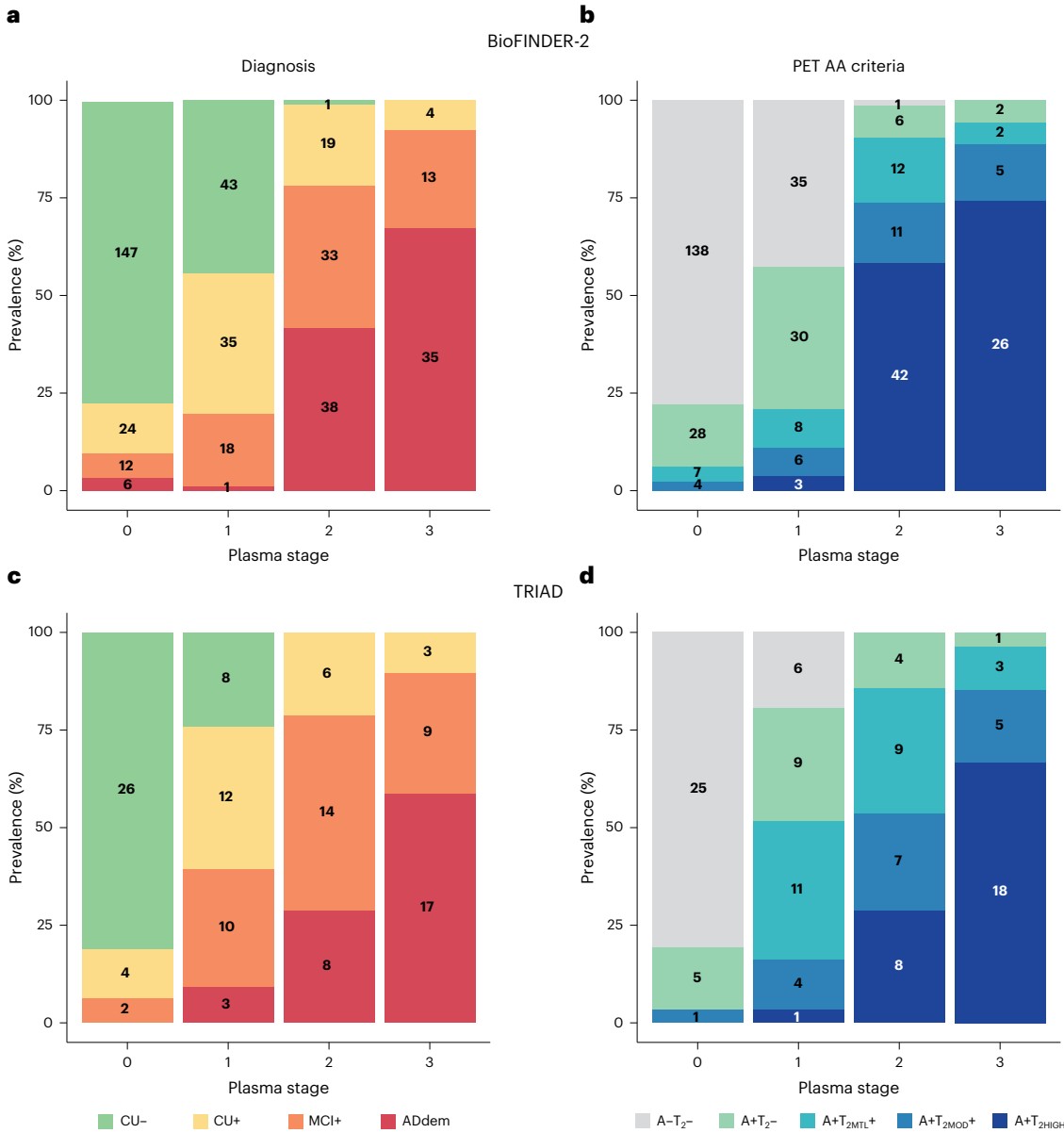

**Fig. 3 | Diagnostic groups and biological staging with the AA criteria classification by plasma stage within the AD continuum. a–d**, Associations with diagnostic groups (BioFINDER-2 in **a** and TRIAD in **c**) and with AA criteria (BioFINDER-2 in **b** and TRIAD in **d**). The number of individuals in each group per stage is detailed in the barplots. Individuals with non-AD diagnosis were excluded from these analyses.

$\beta_{2vs1} = -0.19$ ($-0.24$ to $-0.13$), $P < 0.001$; and $\beta_{3vs2} = -0.17$ ($-0.25$ to $-0.09$), $P < 0.001$. Longitudinal cognitive decline, assessed using mPACC and MMSE scores, demonstrated that higher plasma stages were associated with faster rates of cognitive deterioration in both measures. For mPACC, significant differences were found between successive plasma stages: stage 0 versus stage 1 ($\beta = -0.11$ ($-0.18$ to $-0.03$), $P = 0.009$), stage 1 versus stage 2 ($\beta = -0.42$ ($-0.53$ to $-0.32$), $P < 0.001$) and stage 2 versus stage 3 ($\beta = -0.19$ ($-0.34$ to $-0.04$), $P = 0.014$). For MMSE, significant differences were observed between stage 1 and stage 2 ($\beta = -0.50$ ($-0.60$ to $-0.41$), $P < 0.001$).

## Discussion

In this study, we present a blood-based biomarker staging model for AD using plasma tau biomarkers measured in a single-run MS analysis of a single sample. Our results demonstrate that different plasma tau biomarkers can be used to stage AD biologically in a clinically relevant manner, as we previously showed in CSF[22]. Specifically, we developed a four-stage model based on p-tau217r, p-tau205r and 0N tau.

This model was created and tested in one cohort and validated in an independent cohort, showing strong consistency in the distribution of amyloid-PET, tau-PET and neurodegeneration biomarkers, as well as cognitive outcomes, across stages. Furthermore, blood-based stages proved valuable in detecting clinical diagnostic and PET-based stages, as well as associating with the longitudinal progression of the disease, including Aβ- and tau-PET accumulation, atrophy and cognitive decline. Overall, our findings support the utility of distinct blood tau biomarkers for staging AD, offering significant potential for use in both clinical practice and trials.

Previous studies on brain and CSF have suggested that changes in specific phosphorylated and nonphosphorylated tau species occur at distinct moments across AD[9,11]. We first assessed how plasma tau species change along the AD continuum in relation to clinical diagnosis and Aβ status. Our results demonstrate that different plasma tau biomarkers become abnormal at distinct phases of the disease progression. Plasma p-tau217, alongside p-tau231, showed early elevations in cognitively unimpaired individuals transitioning to Aβ positivity.

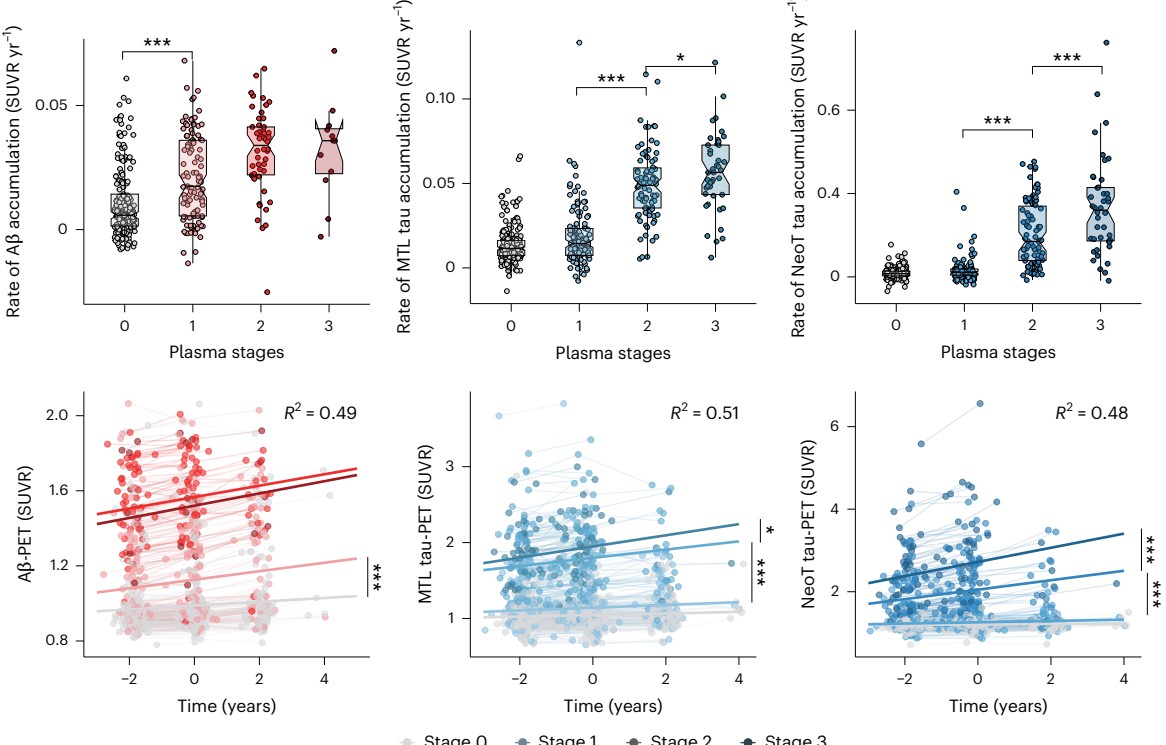

**Fig. 4 | Longitudinal trajectories of AD pathology imaging biomarkers by plasma stage in the BioFINDER-2 cohort.** Individual slopes of each biomarker (from unadjusted linear mixed models) are shown in the first row by plasma stage, while actual longitudinal trajectories, colored by plasma stage, are shown in the second row. Colored thick lines represent the mean trajectory per each plasma stage group as calculated with a linear mixed-effect model adjusting for age, sex and *APOE* ε4 carriership. Statistically significant differences ($P < 0.05$, two-sided) on group slopes were extracted from linear mixed models and show differences only by contiguous groups. Boxplots summarize data distribution, showing the median (central line), interquartile range (IQR; box) and whiskers extending to 1.5× the IQR. Plasma draw was considered time 0 for these analyses. Imaging biomarkers were included from all time points available (including before blood draw). We included all individuals with longitudinal data available for each analysis (Aβ-PET: $n = 368$, MTL tau-PET and NeoT tau-PET: $n = 480$). Exact $P$ values can be found in Supplementary Table 7. *$P < 0.05$, ***$P < 0.001$.

However, p-tau217 exhibited the largest fold changes. Plasma p-tau205 increased later, closer to the onset of symptoms, suggesting it may serve as a more sensitive marker for more advanced disease and accompanying cognitive decline. Nonphosphorylated 0N-tau reached abnormal levels at AD dementia. These findings align with previous studies that identify p-tau217 as the most sensitive plasma biomarker for detecting AD, marked by substantial fold changes and significant longitudinal shifts[12,25,34–37]. Moreover, while primarily observed in CSF[14,15,38] and to a lesser extent in plasma studies[30,31], p-tau205 has been more strongly associated with tau-PET imaging and cognitive decline. In addition, brain studies have shown that 0N tau, along with 4R tau, are more prone to aggregation than other isoforms and can be detected in insoluble tau extracts[9,10]. Conversely, not all tau species—such as p-tau199, p-tau202 or the nonphosphorylated tau variants—showed abnormal changes as the disease progressed. Notably, we report that blood levels of the PNS tau isoform remain unaffected by AD progression. Recent research indicates that PNS tau—also known as big tau—does not undergo typical AD-related hyperphosphorylation, is more efficiently ubiquitinated and degraded, has enhanced microtubule-binding capacity and shows a reduced tendency for aggregation[39]. In that study, patients with AD displayed elevated levels of big tau in the pathology-resistant cerebellum, but not in the pathology-sensitive cortex. However, we did not observe such elevated levels in the blood of patients with AD.

Based on these findings, we developed a staging model utilizing p-tau217r, p-tau205r and 0N tau. For p-tau217 and p-tau205, we used the ratios of phosphorylated to nonphosphorylated peptides (tau212-221 and tau195-209, respectively), as we show that these ratios can mitigate the impact of elevated tau levels in individuals with chronic kidney dysfunction, corroborating previous results[33,40], and had greater consistency across cohorts. The model was also tested using concentrations instead of MS ratios, as they may offer greater applicability and translational potential in clinical settings without greatly diminishing accuracy[41,42]. We identified a consistent pattern of change in Aβ and tau-PET binding, cortical thickness and cognitive decline across cohorts and plasma stages. Aβ-PET primarily exhibited changes between stages 0 and 1, while tau-PET increases were observed between stages 1 and 3. Neurodegeneration markers and cognitive function were characterized by a continuous decrease with advancing stages. This consistency across cohorts was also evident when using plasma stages to predict clinical diagnosis based on cognitive performance and Aβ status. Individuals in plasma stage 0 were mostly cognitively unimpaired, while those in stage 1 were either cognitively unimpaired or MCI. By stage 2, 100% of individuals were Aβ+, with an increased likelihood of cognitive impairment, which further escalated in stage 3. This staging model could serve as a valuable first-line screening tool that, when assessed by a skilled examiner, may complement clinical cognitive diagnosis and improve patient management in clinical settings.

An even more valuable application of this staging system may lie in the capability to detect tau load and spreading in the brain, making it particularly useful for clinical trials and therapeutic interventions. While successful Aβ-targeting immunotherapies have been shown to slow cognitive decline by 25–40% (refs. 4,43), their efficacy is reduced in patients with high tau-PET burden[4,5]. A recent study demonstrated that using plasma p-tau217 alone could reduce the number of participants needed for a clinical trial by 75%, and by 94% when combined with tau-PET imaging[44]. This highlights the potential of incorporating

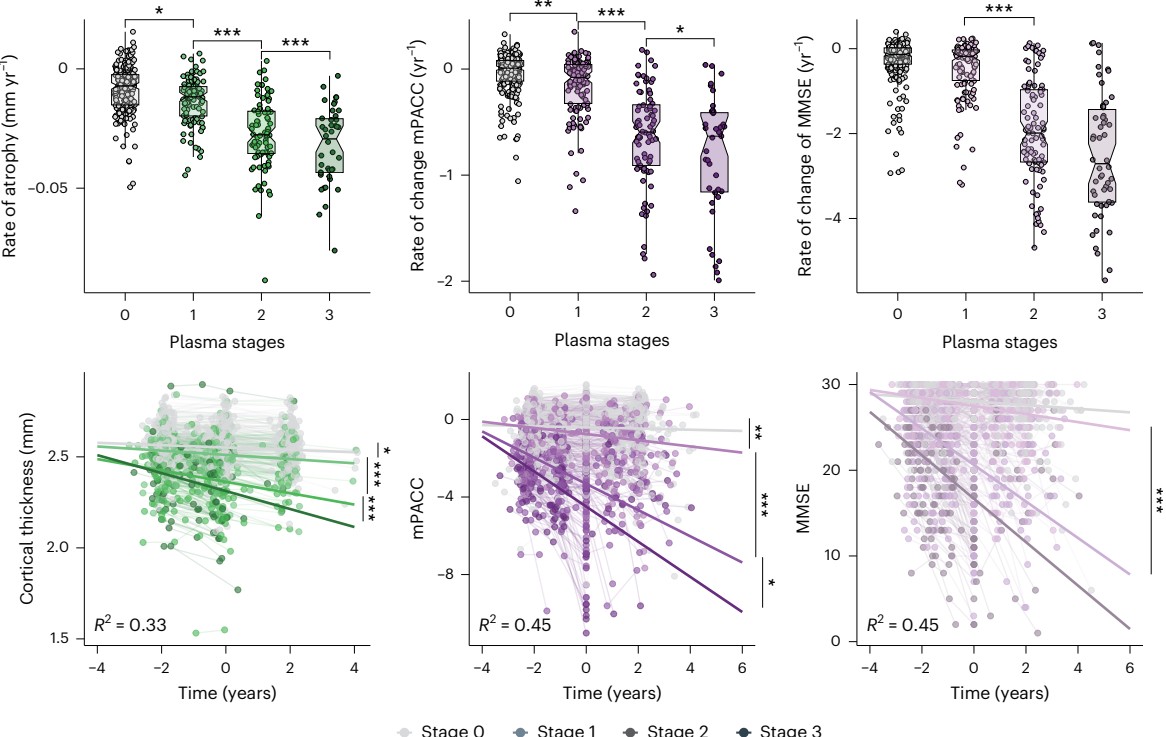

**Fig. 5 | Longitudinal trajectories of atrophy and cognitive decline by plasma stage in the BioFINDER-2 cohort.** Individual slopes of each biomarker (from unadjusted linear mixed models) are shown in the first row by plasma stage, while actual longitudinal trajectories, colored by plasma stage, are shown in the second row. Colored thick lines represent the mean trajectory per each stage group as calculated with a linear mixed-effect model adjusting for age, sex and *APOE* ε4 carriership (and years of education for cognitive outcomes). Statistically significant differences (*P* < 0.05, two-sided) on group slopes were extracted from linear mixed models and show differences only by contiguous groups.

Individuals with non-AD diagnosis were excluded from these analyses to avoid bias. Boxplots summarize data distribution, showing the median (central line), interquartile range (IQR; box) and whiskers extending to 1.5× the IQR. Plasma draw was considered time 0 for these analyses. Atrophy and cognitive measures were included from all time points available (including before blood draw). We included all individuals with longitudinal data available for each analysis (thickness: *n* = 457, mPACC: *n* = 460 and MMSE: *n* = 491). Exact *P* values can be found in Supplementary Table 7. **P* < 0.05, ***P* < 0.005, ****P* < 0.001.

plasma p-tau217 into clinical trial workflows to dramatically reduce the need for tau-PET scans, improving efficiency and cutting costs. However, despite plasma p-tau217's strong ability to identify individuals with elevated tau-PET uptake, its ability to predict continuous tau-PET load is limited[45–48]. By combining biomarkers within a fluid staging system, the process could be further refined, streamlining eligibility workflows and potentially reducing the number of PET scans even further, thus enhancing the overall trial design. In previous studies, we demonstrated that different plasma tau biomarkers are associated with Aβ- and tau-PET imaging[30] and neuropathological examination[31] in distinct ways. In this study, we observed a correlation between plasma tau stages and PET-based stages. By plasma stage 0, most individuals were tau-PET negative; in stage 1, most were either tau-PET negative or in early tau-PET stages; by stage 2, individuals were primarily in intermediate stages, and by stage 3, the majority were in advanced stages. However, plasma stages did not perfectly align with imaging stages, which was highlighted as a possibility by the new AA criteria[27,28]. Plasma stages appear to precede imaging stages, which could be explained by two factors: the sensitivity of tau-PET tracers and the differences in the biomarkers being measured. On the one hand, post-mortem studies have shown that a positive [¹⁸F]flortaucipir tau-PET scan (whether assessed quantitatively or visually) typically reflects tau pathology at Braak stage IV or higher[49], indicating that the method is not sensitive for detecting very early tau pathology. On the other hand, plasma measures capture soluble tau forms, whereas tau-PET tracers bind to insoluble tau aggregates in the brain[18]. These factors may also indicate that fluid stages are more sensitive in detecting early tau pathology.

This is supported by recent data showing that tau-PET outperforms plasma p-tau217 in predicting cognitive decline in symptomatic individuals[50] but is not significantly higher than plasma p-tau217 in Aβ-negative and Aβ-positive cognitively unimpaired individuals[44]. Further research is required to fully understand the relationship and differences between plasma tau stages and tau imaging stages.

Another significant benefit of an easily accessible staging system lies in its predictive power to determine the expected progression of AD. Our plasma staging model was associated with longitudinal tau-PET accumulation. Individuals classified as stage 0 or stage 1 showed no evidence of an increase in tau-PET signal over time, whereas those in stages 2 and 3 demonstrated significant increases. Interestingly, the rate of tau-PET signal increases in the medial temporal lobe was similar between stages 2 and 3, whereas neocortical temporal regions exhibited more pronounced differences in tau-PET uptake between these stages. This suggests that individuals in stage 2 will continue to accumulate tau aggregates in both brain regions, while those in stage 3 may have reached the maximum accumulation in the medial temporal lobe but will continue to accumulate tau in later accumulation areas. This information could be crucial for identifying patients who may benefit from treatment, especially in the context of future anti-tau therapies, where predicting tau spread in the brain will be critical. In addition, each plasma stage was associated with a different mean rate of cognitive decline. This capability could assist clinicians to forecast individual deterioration based on plasma stage. For example, the efficacy of anti-amyloid therapies to delay cognitive decline has been suggested to be affected not only by tau-PET

levels, but also cognitive status. Individuals with same low-to-medium tau-PET burden but more advanced clinical dementia ratings may have fewer additional months of independence with the same treatment[51].

The main strengths of this study include the simultaneous measurement of multiple plasma tau species in a single analysis, allowing us to identify which biomarkers are most effective for staging AD and to develop a staging system that reflects various pathological changes. In addition, this model was validated across two independent cohorts, with the inclusion of longitudinal data further supporting its robustness. However, this study is not without limitations. We observed a late increase in the levels of p-tau181 compared with previous reports, probably due to differences in the specific target measured between MS and immunoassay-based methods, as described[30]. In addition, while biofluid concentrations represent a snapshot of the balance between production and clearance of specific proteins at a given time, imaging biomarkers reflect the cumulative aggregation of pathology over time. As a result, direct comparisons between fluid and imaging biomarkers may not always be precise. Moreover, PET imaging acquisition in the two cohorts was conducted using different methodologies (that is, scanners and tracers), which may have influenced the classification of PET status and, consequently, the relationship between imaging stages and plasma stages. In addition, because the blood data were not acquired at baseline but at the second visit of the BioFINDER-2 cohort, the available follow-up data were insufficient to perform purely prospective analyses. Therefore, we also included retrospective data in our longitudinal analyses. In this regard, time-to-cognitive decline analysis could not be performed due to the limited availability of prospective longitudinal data. We acknowledge the potential limitations and biases introduced by this approach, and further research is needed to validate these findings. Furthermore, the results were evaluated at the group level, limiting direct clinical application. In this regard, while the staging system may be effective at the group level, further work is needed to improve its specificity for potential clinical use. Future studies should validate the staging model in cohorts with post-mortem neuropathological examination and assess whether incorporating additional blood biomarkers could enhance the accuracy of the plasma staging system. Moreover, research involving more diverse cohorts, particularly those with a focus on peripheral comorbidities, is needed to evaluate how these might influence the model. Finally, standardization of the staging model across various settings—through the establishment of cutoffs, common reference materials and validation using different technologies—will be essential to advance this model toward clinical implementation.

In conclusion, we have developed and validated a biomarker-based staging model using plasma tau biomarkers from a single sample and analysis. Our results demonstrate that distinct plasma tau biomarkers can accurately stage AD, offering valuable insights for identifying clinical diagnostic and imaging stages, as well as predicting longitudinal tau-PET accumulation, neurodegeneration and cognitive decline. We believe these findings will enhance the potential application of blood tau biomarkers, improving patient management in both clinical trials and routine clinical practice.

## Methods

### Participants

**BioFINDER.** This study included participants from the BioFINDER-2 (NCT03174938) cohort covering the whole AD continuum from preclinical disease to dementia. Cognitively unimpaired individuals were recruited from a population-based study in Malmö, Sweden, as previously described[52]. These participants did not meet the criteria for mild cognitive impairment (MCI) or dementia at baseline. In addition, patients with subjective cognitive decline, MCI and AD dementia were recruited from the memory clinics of Skåne University Hospital and Ängelholm hospital in Sweden. All participants provided written informed consent and were recruited consecutively without excluding any eligible participants between March 2019 and November 2022. MCI was defined using a broad neuropsychological battery and dementia according to the Diagnostic and Statistical Manual of Mental Disorders (DSM-5) criteria for major neurocognitive disorder, as previously described[52,53]. Patients with MCI or dementia were required to be Aβ positive—determined by either CSF or PET—to be considered as having AD. Participants with other etiological diagnosis than AD were classified as non-AD and are referred as cognitively impaired not due to AD. General cognition was assessed using the MMSE and memory using the ten-word delayed recall test from the Alzheimer's Disease Assessment Scale-Cognitive subscale[54]. A modified preclinical Alzheimer cognitive composite was used to assess cognitive decline sensitive to the earliest changes[55], as previously described[22]. The study was approved by the Regional Ethics Committee in Lund, Sweden (Dnr 2016-1053).

**TRIAD.** This study assessed individuals from the TRIAD[56] cohort. TRIAD patients were approached consecutively, and no eligible participants were excluded. Data from the TRIAD study were collected between October 2019 and May 2024. All participants provided written informed consent and had clinical evaluations by dementia specialists, plasma assessments and structural magnetic resonance imaging (MRI), as well as amyloid-PET with [18F]AZD4694 and tau-PET with [18F]MK6240. Evaluations of participants included a review of their medical history and an interview with the participant and their study partner, a neurological examination by a dementia specialist and a detailed neuropsychological examination. Participants were approached consecutively, and data were collected prospectively. Cognitively unimpaired individuals had no objective cognitive impairment and a CDR score of 0. Individuals with MCI had subjective and/or objective cognitive impairment and a CDR score of 0.5. Individuals with dementia had a CDR 1 or greater. Patients with MCI or dementia had to be Aβ positive by either CSF or PET to be considered due to AD. Patients with cognitive impairment but negative Aβ status were classified as cognitively impaired not due to AD. Participants were excluded from this study if they had systemic conditions that were not adequately controlled through a stable medication regimen. Other exclusion criteria were active substance abuse, recent head trauma, recent major surgery and MRI and PET safety contraindications. The study was approved by the Montreal Neurological Institute PET working committee and the Douglas Mental Health University Institute Research Ethics Board (ethical approval: MP-18-2019-223, IUSMD-19-05).

### Plasma tau MS analysis

Available plasma samples were analyzed by MS detection of phosphorylated and nonphosphorylated tau peptides following a previously established protocol[21,30]. In summary, 1 ml of plasma samples (collected with EDTA) were thawed, vortexed at 2,000 rpm for 30 s and centrifuged for 10 min at 4,000g. Tau protein was isolated via immunoprecipitation with a combination of three antibodies (Tau12, purified anti-tau, 6–18 antibody, 806501, BioLegend; HT7, tau monoclonal antibody, MN1000, Thermofisher; and BT2, tau monoclonal antibody, MN1010, Thermofisher), followed by enrichment using perchloric acid precipitation and desalting. The samples were then subjected to tryptic digestion by adding trypsin solution (sequencing grade, Promega) at 0.1 µg per sample (final concentration of 2.5 µg ml⁻¹ in 50 mM ammonium bicarbonate) and incubating at 37 °C overnight. After 18 h, the digestion was halted by adding trifluoroacetic acid to a final concentration of 0.1%. The samples were then lyophilized and stored at −20 °C until MS analysis. For MS analysis, a hybrid Orbitrap mass spectrometer (Lumos, Thermo Scientific) equipped with an EasySpray nano-ESI ion source was used. The instrument was operated in positive ion mode with the following settings for parallel reaction monitoring scans: higher-energy collisional dissociation activation, Orbitrap detector, 60,000 resolution, scan range of 250–1,200 $m/z$, radio frequency lens at 30%, and Easy-IC enabled. Quadrupole isolation was applied with a

0.7 *m/z* window. Parameters such as maximum injection time, normalized automatic gain control target, optimal collision energy and field asymmetry ion mobility spectrometry voltage were optimized for each peptide. Details of the endogenous tryptic peptides analyzed are provided in Supplementary Table 8. Liquid chromatography–MS data acquisition was performed using Xcalibur 4.5 and Tune 3.5 software (Thermo Scientific), and the data were analyzed with Skyline (McCoss Lab, University of Washington). The analysis of plasma samples was conducted without knowledge of participant information.

### Imaging acquisition and processing

In BioFINDER-2, imaging procedures have been described previously[22]. In brief, Aβ-PET and tau-PET were acquired using [18F]flutemetamol and [18F]RO948, respectively. Amyloid-PET binding was measured as the standardized uptake value ratio (SUVR) using a neocortical meta-region of interest (ROI) and with whole cerebellum as a reference region. Of note, most of the patients with AD dementia did not undergo amyloid-PET in BioFINDER-2 owing to the study design. Tau-PET binding was measured in regions covering early (MTL) and intermediate (NeoT) tau deposition areas[57]. For assessing cortical thickness, T1-weighted anatomical magnetization-prepared rapid gradient echo (MPRAGE) images (1-mm isotropic voxels) were used. A cortical thickness meta-ROI was calculated including entorhinal, inferior temporal, middle temporal and fusiform using FreeSurfer (version 6.0; https://surfer.nmr.mgh.harvard.edu) parcellation, which are areas known to be susceptible to AD-related atrophy[58]. In TRIAD, [18F]AZD4694 PET and [18F]MK6240 PET scans were acquired with a brain-dedicated Siemens High Resolution Research Tomograph. [18F]AZD4694 PET images were acquired 40–70 min after bolus injection and reconstructed on a four-dimensional volume with three frames (3 × 600 s), as previously described[59]. [18F]MK6240 PET images were acquired at 90–110 min after bolus radiotracer injection and reconstructed on a four-dimensional volume with four frames (4 × 300 s) (ref. 60). A 6-min transmission scan with a rotating 137Cs point source followed each PET acquisition for attenuation correction. PET images were corrected for decay, motion, dead time, random and scattered coincidences. T1-weighted MRIs were acquired at the Montreal Neurological Institute on a 3 T Siemens Magnetom using a standard head coil. They underwent correction for nonuniformity and field distortion and were processed using an in-house pipeline. PET images were automatically registered to the T1-weighted image space, and the T1-weighted images were linearly and nonlinearly registered to the Montreal Neurological Institute (MNI) reference space. To minimize interference of meningeal spillover, [18F]MK6240 images were stripped of meningeal signal in native space before spatial normalization and smoothing[61]. [18F]AZD4694 SUVR maps were calculated using the whole cerebellum gray matter as the reference region, and [18F]MK6240 SUVR maps were generated using the inferior cerebellar gray matter as a reference region. Spatial smoothing allowed the PET images to achieve an 8-mm full-width at half-maximum resolution. Amyloid-β SUVR from a neocortical ROI for each participant was estimated by averaging the SUVR from the precuneus, prefrontal, orbitofrontal, parietal, temporal and cingulate cortices[59], with amyloid-β positivity defined as an [18F]AZD4694 above 1.55 (centiloids = 24). The SUVR from the temporal meta-ROI, a composite mask commonly used as a summary measure of tau-PET, was calculated from the entorhinal, parahippocampal, amygdala, fusiform, inferior and middle temporal cortices, as previously described[62], with positivity defined as SUVRs above 1.24 (ref. 63). T1-weighted MRIs were acquired at the MNI on a 3 T Siemens Magnetom scanner using a standard head coil. A composite neurodegeneration summary measure was used, comprising a surface-area weighted average of the mean cortical thickness in the following individual ROIs bilaterally: entorhinal, inferior temporal, middle temporal and fusiform as previously described[62]. Cortical thickness measures were derived from FreeSurfer (v7.4).

### Statistics and reproducibility

First, we plotted each plasma biomarker against clinical diagnosis using CU− as the control group to create *z* scores of each biomarker. Individuals with non-AD diagnoses were excluded from this analysis. Differences across diagnoses were calculated using analysis of covariance (ANCOVA) adjusted by age and sex and *APOE* ε4 carriership followed by Tukey's corrected, post-hoc pairwise comparisons (two-sided analysis). Those biomarkers that showed differences across the consecutive diagnosis groups were selected for creating the staging model (that is, p-tau217r, p-tau205r and 0N-tau).

**Creation of the model.** Each participant from the BioFINDER-2 cohort was characterized by these three main biomarkers as a point in a three-dimensional space, in which each biomarker corresponded to an orthogonal axis (Extended Data Fig. 2). We then divided the whole BioFINDER-2 cohort into ten partitions (or ten folds). We used 9/10 parts of the sample to train a *k*-means clustering approach that aimed to divide the sample into four groups (or stages) using the clusterboot function (fpc package, v 2.2-11). We applied the resulting model to the test data (1/10 of the BioFINDER-2 cohort) and the validation set (TRIAD cohort). This was repeated for each of the ten parts of the BioFINDER-2 cohort, so the model was created in a completely independent sample from the one it was tested. This approach was repeated five times with different initialization parameters to avoid randomness in the model creation and application. The final stage for each individual was assessed as the mode of the five repetitions in the BioFINDER-2 individuals and all the classifications (50) of the TRIAD participants. For testing replicability, we looked at how many times each individual was classified in the same stage with 5,000 different initializations.

**Cross-sectional assessment of the plasma staging model.** Next, we compared the groups created with the clustering approach to other biomarkers and disease scales cross-sectionally. Statistical differences by plasma stage were assessed using ANCOVA adjusted by age and sex and *APOE* ε4 carriership and years of education for cognitive outcomes followed by Tukey's corrected, post-hoc pairwise comparisons (two-sided analysis). These analyses were performed for BioFINDER-2 and TRIAD participants independently.

**Longitudinal assessment of the plasma staging model.** Finally, we also performed longitudinal analyses to assess tau-PET accumulation, atrophy and cognitive decline in the BioFINDER-2 cohort. To this aim, we used linear mixed models (lme4 package, v1.1-34) with tau, cognition or cortical thickness as the outcome and the interaction between time and plasma stages as the predictor, with random slopes and intercepts adjusting for age and sex and *APOE* ε4 carriership (and years of education for cognitive outcomes). Importantly, all available outcome data were used to calculate these changes, including data from before and after the acquisition of the plasma samples (−2 to 4 years before/after plasma sampling).

All analyses were performed with R (version 4.3.1). Two-sided *P* values less than 0.05 were considered statistically significant. For comparisons between plasma stages (that is, biomarker levels and rates of change), false discovery rate correction was applied to account for multiple comparisons. All plots were done with the ggplot2 package (v3.5.1).

### Reporting summary

Further information on research design is available in the Nature Portfolio Reporting Summary linked to this article.

## Data availability

This study includes no data deposited in external repositories. Anonymized data can be shared upon reasonable request from a qualified academic investigator, for the sole purpose of replicating

procedures and results presented in the Article, as long as data transfer agrees with local legislation and with the local ethical review board of each cohort, which must be regulated in a material/data transfer agreement. Researchers interested in accessing the datasets should contact the corresponding authors and provide a brief research proposal outlining the intended use of the data. Data requests will be evaluated on the basis of scientific merit and compliance with ethical and legal requirements. Cohort-specific guidelines: BioFINDER-2 (https://biofinder.se) and TRIAD (https://triad.tnl-mcgill.com).

## Code availability

The code used for the analyses in this study is available from the corresponding authors upon request. Interested researchers are encouraged to contact the corresponding authors with a brief description of the intended use. Access will be provided for noncommercial purposes and solely for reproducing the results presented in this Article.

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

## Acknowledgements

The BioFINDER-2 study (O.H.) was supported by the National Institute on Aging (R01AG083740), European Research Council (ADG-101096455), Alzheimer's Association (ZEN24-1069572, SG-23-1061717), GHR Foundation, Swedish Research Council (2018-02052, 2021-02219 and 2022-00775), ERA PerMed (ERAPERMED2021-184), Knut and Alice Wallenberg foundation (2022-0231), Strategic Research Area MultiPark (Multidisciplinary Research in Parkinson's disease) at Lund University, Swedish Alzheimer's Foundation (AF-980907 and AF-994229), Swedish Brain Foundation (FO2021-0293 and FO2023-0163), Parkinson Foundation of Sweden (1412/22), Cure Alzheimer's fund, Rönström Family Foundation, Konung Gustaf V:s och Drottning Victorias Frimurarestiftelse, Skåne University Hospital Foundation (2020-O000028), WASP and DDLS joint call for research projects (WASP/DDLS22-066), Lilly Research Award Program, Regionalt Forskningsstöd (2022-1259), the Bundy Academy, and Swedish federal government under the ALF agreement (2022-Projekt0080 and 2022-Projekt0107). The precursor of 18F-flutemetamol was sponsored by GE Healthcare. The precursor of 18F-RO948 was provided by Roche. The MS measurements (K.B. and H.Z.) were supported by grants from the Swedish Research Council (#2023-00356, #2022-01018, #2019-02397, #2017-00915 and #2022-00732), the European Union's Horizon Europe research and innovation program under grant agreement no. 101053962, Swedish State Support for Clinical Research (#ALFGBG-71320), the AD Strategic Fund and the Alzheimer's Association (#ADSF-21-831376-C, #ADSF-21-831381-C, #ADSF-21-831377-C and #ADSF-24-1284328-C), the BrightFocus Foundation (A2022015F), the Swedish Dementia Foundation, Gun and Bertil Stohnes Foundation, Åhlén-stiftelsen, Alzheimerfonden (AF-968621) and Gamla Tjänarinnor Foundation, and Hjärnfonden, Sweden (#ALZ2022-0006 and #FO2024-0048-TK-130), the Swedish state under the agreement between the Swedish government and the County Councils, the ALF agreement (#ALFGBG-965240 and #ALFGBG-1006418). The TRIAD (P.R.-N.) cohort is supported by the Weston Brain Institute, Canadian Institutes of Health Research (MOP-11-51-31; RFN 152985, 159815 and 162303), Canadian Consortium on Neurodegeneration in Aging (MOP-11-51-31-team 1), the Alzheimer's Association (NIRG-12-92090 and NIRP-12-259245), Brain Canada Foundation (CFI Project 34874; 33397), the Fonds de Recherche du Québec–Santé (Chercheur Boursier, 2020-VICO-279314) and the Colin J. Adair Charitable Foundation. The funding sources had no role in the design and conduct of the study; in the collection, analysis or interpretation of the data; or in the preparation, review or approval of the manuscript. The authors received no specific funding for this work.

## Author contributions

L.M.-G., G.S., K.B. and O.H. created the concept and design. Data acquisition and analysis was performed by L.M.-G., G.S., J.N., S.W., G.B. and J.G. J.T., S.J., N.J.A., A.L.B., N.R., J.L.-R., N.M.-C., S.P., E.S., H.Z. and P.R.-N. contributed to the sample selection and interpretation of the data. L.M.-G. and G.S. verified the underlying data. L.M.-G., G.S., K.B. and O.H. drafted the manuscript, and all authors revised. All authors read and approved of the final manuscript.

## Funding

## Competing interests

L.M.-G. has received speaker fees from Quanterix and Esteve, and served as consultant for Quanterix. G.S. has received speaker fees from Springer Nature, Esteve, GE Healthcare and Adium and has participated in advisory board for Johnson & Johnson. H.Z. has served at scientific advisory boards and/or as a consultant for AbbVie, Acumen, Alector, Alzinova, ALZpath, Amylyx, Annexon, Apellis, Artery Therapeutics, AZTherapies, Cognito Therapeutics, CogRx, Denali, Eisai, Enigma, LabCorp, Merry Life, Nervgen, Novo Nordisk, Optoceutics, Passage Bio, Pinteon Therapeutics, Prothena, Quanterix, Red Abbey Labs, reMYND, Roche, Samumed, Siemens Healthineers, Triplet Therapeutics and Wave; has given lectures sponsored by Alzecure, BioArctic, Biogen, Cellectricon, Fujirebio, Lilly, Novo Nordisk, Roche and WebMD; and is a co-founder of Brain Biomarker Solutions in Gothenburg AB (BBS), which is a part of the GU Ventures Incubator Program (outside submitted work). S.P. has acquired research support (for the institution) from Avid and ki elements through ADDF. In the past 2 years, he has received consultancy/speaker fees from BioArtic, Biogen, Esai, Eli Lilly, Novo Nordisk and Roche. K.B. has served as a consultant, at advisory boards, or at data monitoring committees for Abcam, Axon, BioArctic, Biogen, JOMDD/Shimadzu, Julius Clinical, Lilly, MagQu, Novartis, Ono Pharma, Pharmatrophix, Prothena, Roche Diagnostics and Siemens Healthineers and is a co-founder of Brain Biomarker Solutions in Gothenburg AB (BBS), which is a part of the GU Ventures Incubator Program, outside the work presented in this Article. P.R.-N. participated in advisory board for Roche, Novo Nordisk and Cerveau (outside submitted work). O.H. is an employee of Eli Lilly and Lund University, and he has previously acquired research support (for Lund University) from Avid Radiopharmaceuticals, Biogen, C2N Diagnostics, Eli Lilly, Eisai, Fujirebio, GE Healthcare and Roche. In the past 2 years, he has received consultancy/speaker fees from ALZpath, BioArctic, Biogen, Bristol Myers Squibb, Eisai, Eli Lilly, Fujirebio, Merck, Novartis, Novo Nordisk, Roche, Sanofi and Siemens. J.T. has served as a paid consultant for Neurotorium and for Alzheon Inc. P.R.-N. participated in advisory board for Roche, Novo Nordisk and Cerveau (outside submitted work). N.M.-C. has received consultancy/speaker fees from Biogen, Merck and Owkin. The other authors declare no competing interests.

## Additional information

**Extended data** is available for this paper at https://doi.org/10.1038/s43587-025-00951-w.

**Correspondence and requests for materials** should be addressed to Laia Montoliu-Gaya or Oskar Hansson.

¹Department of Psychiatry and Neurochemistry, Institute of Neuroscience & Physiology, The Sahlgrenska Academy at the University of Gothenburg, Mölndal, Sweden. ²Clinical Memory Research Unit, Department of Clinical Sciences Malmö, Lund University, Lund, Sweden. ³Translational Neuroimaging Laboratory, McGill Research Centre for Studies in Aging, McGill University, Montreal, Quebec, Canada. ⁴Department of Neurology and Neurosurgery, Faculty of Medicine, McGill University, Montreal, Quebec, Canada. ⁵Banner Sun Health Research Institute, Sun City, AZ, USA. ⁶Banner Alzheimer's Institute, Phoenix, AZ, USA. ⁷Wallenberg Center for Molecular Medicine, Lund University, Lund, Sweden. ⁸Memory Clinic, Skåne University Hospital, Malmö, Sweden. ⁹Clinical Neurochemistry Laboratory, Sahlgrenska University Hospital, Mölndal, Sweden. ¹⁰Department of Neurodegenerative Disease, Queen Square Institute of Neurology, University College London, London, UK. ¹¹UK Dementia Research Institute, University College London, London, UK. ¹²Hong Kong Center for Neurodegenerative Diseases, Hong Kong, China. ¹³UW Department of Medicine, School of Medicine and Public Health, Madison, WI, USA. ¹⁴Paris Brain Institute, ICM, Pitié-Salpêtrière Hospital, Sorbonne University, Paris, France. ¹⁵Neurodegenerative Disorder Research Center, Division of Life Sciences and Medicine, and Department of Neurology, Institute on Aging and Brain Disorders, University of Science and Technology of China and First Affiliated Hospital of USTC, Hefei, China. ¹⁶These authors contributed equally: Laia Montoliu-Gaya, Gemma Salvadó. ¹⁷These authors jointly supervised this work: Kaj Blennow, Oskar Hansson. ✉e-mail: laia.montoliu.gaya@gu.se; oskar.hansson@med.lu.se

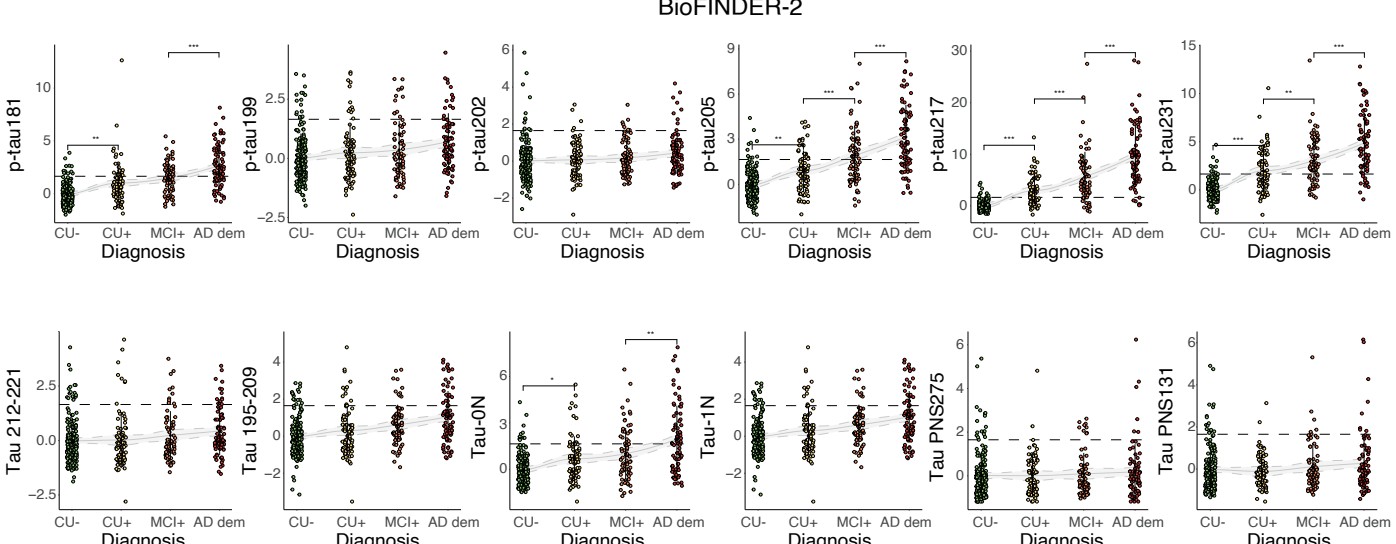

BioFINDER-2

**Extended Data Fig. 1 | Plasma tau biomarkers levels by diagnostic and Aβ status.** Levels were z-scored using CU- group as a reference. Horizontal dashed line is set at z = 1.96, which represents the abnormality (95%CI of CU- group). Individuals with non-AD diagnosis were not included in this analysis. Statistical differences by diagnostic groups were assessed using ANCOVA adjusted by age and sex and *APOE ε*4 carriership followed by Tukey's corrected, post hoc pairwise comparisons (two-sided analysis). In the figure, differences only differences by contiguous groups are shown. Mean (black dot) and SD (vertical black line) plasma levels by diagnostic group are shown. Exact p-values can be found in Supplementary Table 1.*: p < 0.05; **: p < 0.005; ***: p < 0.001. Abbreviations: CU-, cognitively unimpaired Aβ-negative; CU + , cognitively unimpaired Aβ-positive; FDR, false discovery rate; MCI + , mild cognitive impairment Aβ-positive; ADdem, Alzheimer's disease dementia.

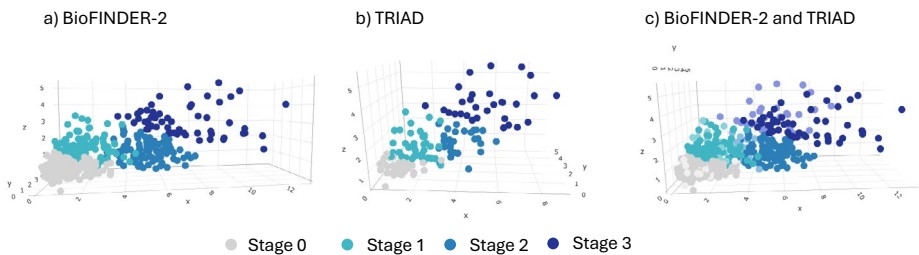

**Extended Data Fig. 2 | Clustering classification of plasma stages.** 3D-plot including the three biomarkers used for the clustering approach (x-axis: p-tau217r, y-axis: p-tau205r and z-axis: Tau-0N) for BioFINDER-2 (**a**), TRIAD (**b**), and both BioFINDER-2 and TRIAD (**c**). Colors represent the plasma stages as classified by the clustering approach. In **c**, TRIAD individuals are shown in the same color but with lighter shade in comparison to BioFINDER-2 individuals.

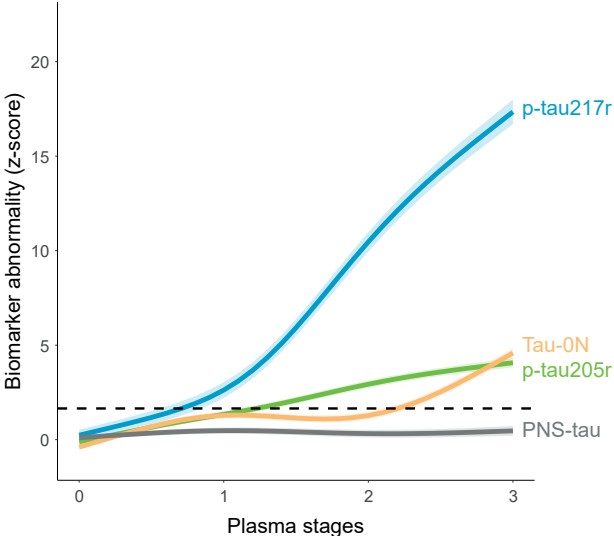

**Extended Data Fig. 3 | Selected tau biomarker trajectories by plasma stages in the BioFINDER-2 cohort.** LOESS regression of the plasma biomarkers included in the staging model plus plasma PNS-tau for comparison. Biomarker levels are z-scored based on a group of CU− participants, and all increases represent an increase in abnormality. Colored lines and bands represent the LOESS regression and its 95% CI. Horizontal line is drawn at z-score = 1.96, which represents 95% CI of the reference group (CU−). Abbreviation: LOESS, locally estimated scatterplot smoothing.

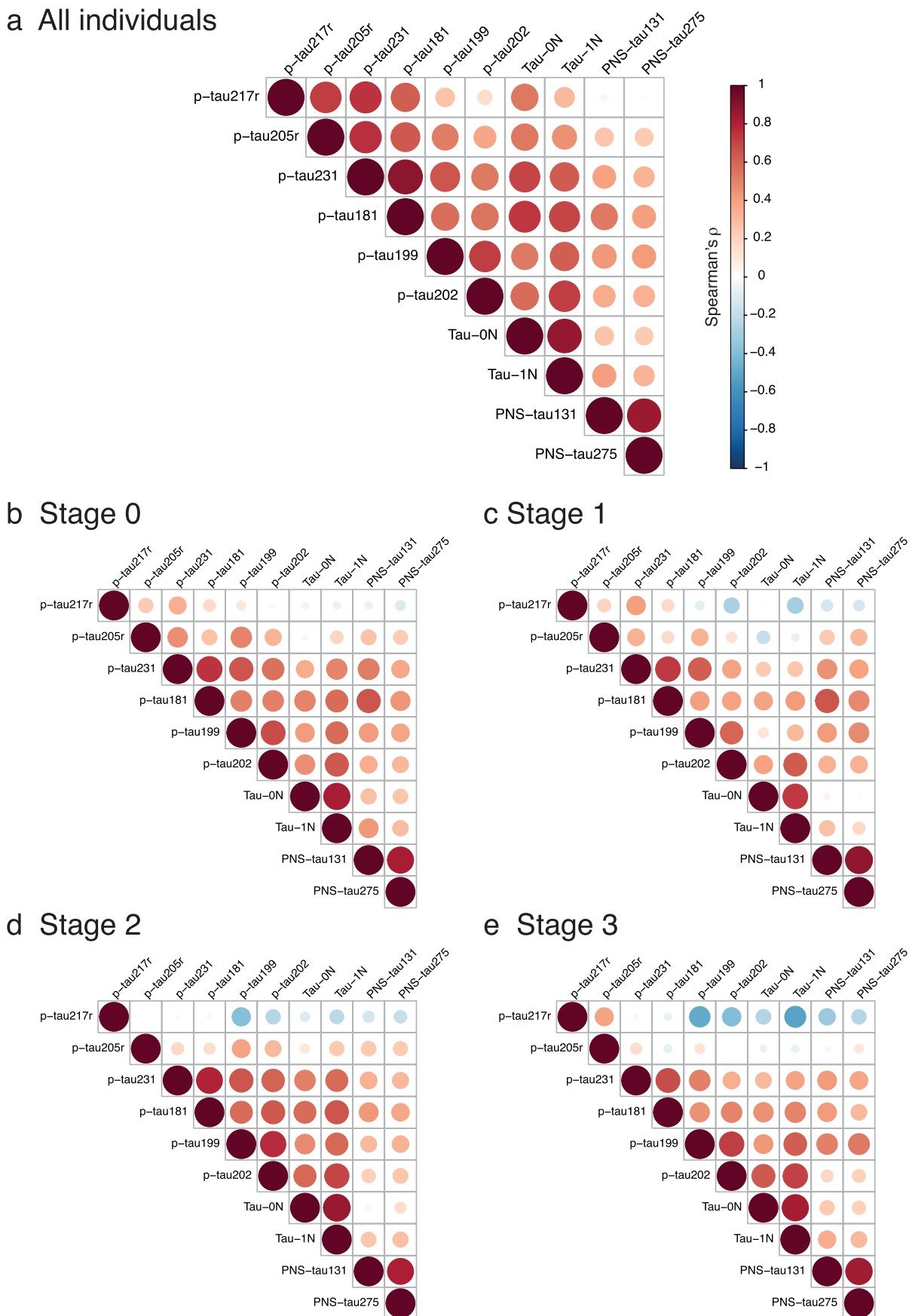

**Extended Data Fig. 4 | Cross-correlation among plasma biomarkers.** Spearman's rho correlations are shown among plasma biomarkers using all (**a**) or by plasma stages participants (**b-e**).

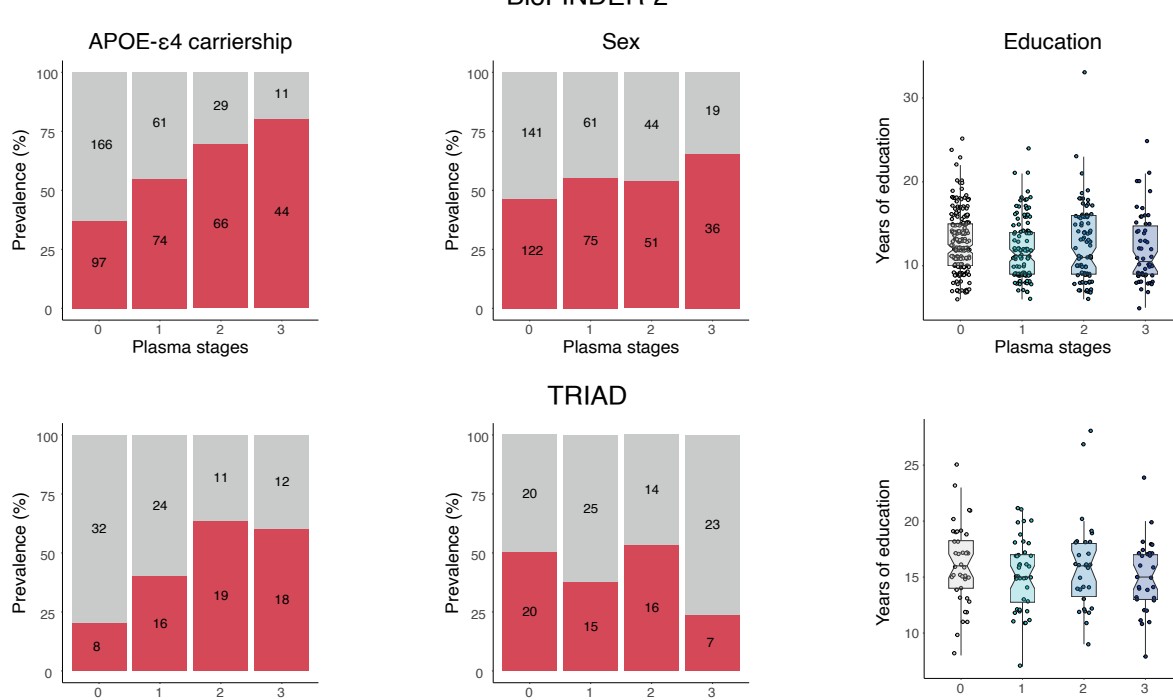

**Extended Data Fig. 5 | Demographic characteristics by plasma stages.**
For *APOE* ε4 carriership and sex, number of individuals in each group per stage are detailed in the barplots (red color represents carriers and women, respectively). Plasma stages were created in the main (BioFINDER-2) cohort and validated in the replication cohort (TRIAD). Years of education by stages defined by plasma biomarkers are also shown. Group comparisons were computed with an ANOVA, followed by Tukey-corrected *post hoc* pairwise comparisons. Only comparison between consecutive groups are shown. Boxplots summarize data distribution, showing the median (central line), interquartile range (IQR; box) and whiskers extending to 1.5 × the IQR. *: p < 0.05; **: p < 0.005; ***: p < 0.001.

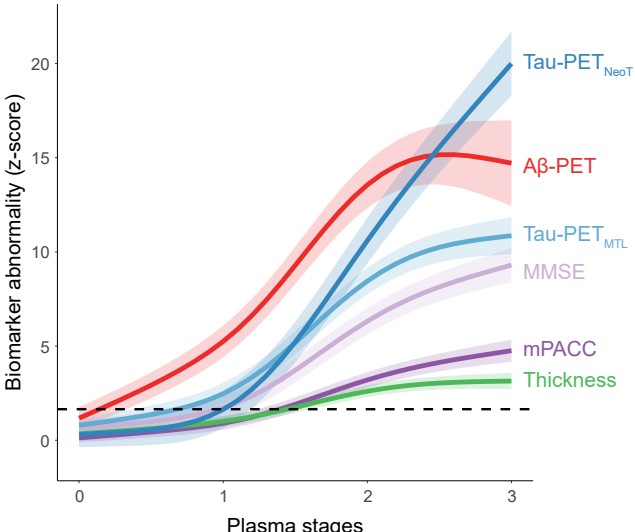

**Extended Data Fig. 6 | AD biomarker trajectories by plasma stages in the BioFINDER-2 cohort.** LOESS regression of imaging and cognitive biomarkers. Biomarker levels are z-scored based on a group of CU− participants, and all increases represent an increase in abnormality. Colored lines and bands represent the LOESS regression and its 95% CI. Horizontal line is drawn at z-score = 1.96, which represents 95% CI of the reference group (CU−). Abbreviations: LOESS, locally estimated scatterplot smoothing. MMSE, Mini-Mental State Examination; mPACC, modified preclinical Alzheimer's cognitive composite; MTL, medial temporal lobe; NeoT: temporal neocortex.

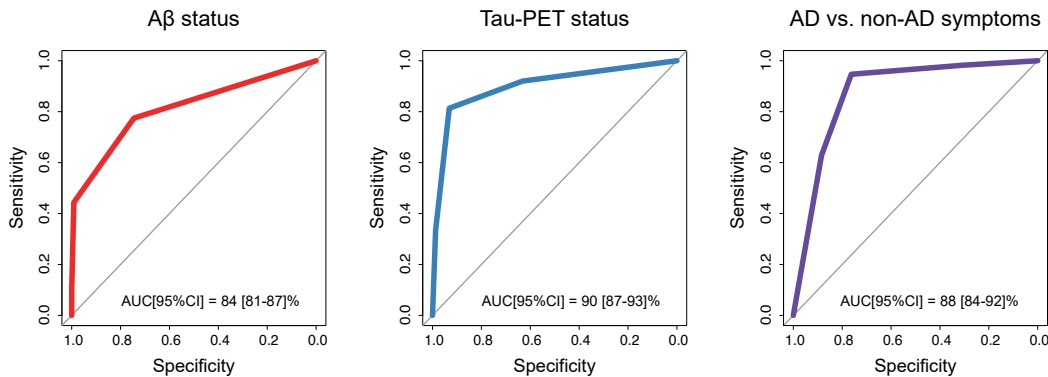

**Extended Data Fig. 7 | Plasma stages for predicting dichotomic categories.** ROC curves were used to assess the classification of Aβ-PET, tau-PET and AD versus non-AD cognitive impairment in the BioFINDER-2 cohort. Abbreviations: AUC, area under the curve; ROC, receiver operating characteristic.

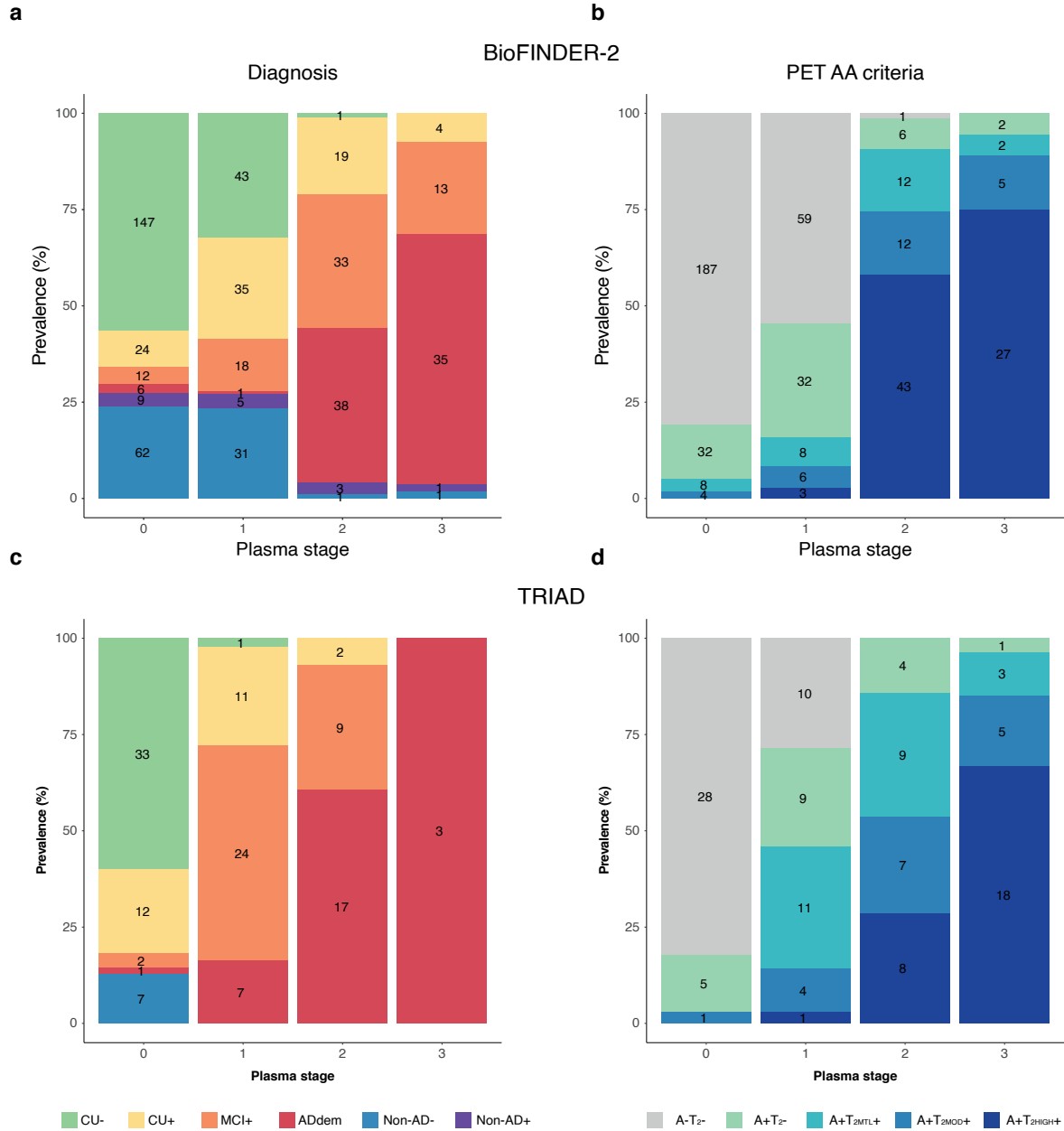

**Extended Data Fig. 8 | Diagnostic groups and biological staging with the AA criteria classification by plasma stages in all participants in the BioFINDER-2 and TRIAD cohorts.** Panels **a** (BioFINDER-2) and **c** (TRIAD) show the association with diagnostic groups, and **b** (BioFINDER-2) and **d** (TRIAD) with the AA criteria. The number of individuals in each group per stage is detailed in the barplots. Only individuals who did not follow the expected biological staging were excluded from PET AA criteria analyses. Abbreviations: AA, Alzheimer's Association; A-T$_2$-, Aβ-negative and tau-negative; A + T$_2$-, Aβ-positive and tau-negative; A + T$_{2MTL+}$, Aβ-positive and tau-PET positive in the medial temporal region; A + T$_{2MOD+}$, Aβ-positive and tau-PET moderate neocortical uptake; A + T$_{2HIGH+}$, Aβ-positive and tau-PET high neocortical uptake; CU-, cognitively unimpaired Aβ-negative; CU + , cognitively unimpaired Aβ-positive; MCI + , mild cognitive impairment Aβ-positive; ADdem, Alzheimer's disease dementia.

