## [Peer Review File · Nature Aging]

PLASMA TAU BIOMARKERS FOR BIOLOGICAL STAGING OF ALZHEIMER'S DISEASE

Corresponding Author: Dr Laia Montoliu-Gaya

Version 0:

Reviewer comments:

Reviewer #1

(Remarks to the Author)

This is a two site discovery and replication exercise of a panel of plasma tau markers in an effort to develop a staging system in the plasma that proxies brain tau abundance and topographic spread. Using ptau217ratio to nonphosphorylated forms, ptau205ratio to nonphosphorylated forms and 0Ntau that were then normalized to their CU group, the authors performed a k-means clustering to create 3 groups. The authors did some further analyses to confirm relative stability of the classification. The key measure the biomarker stage derived from the 3 plasma tau biomarker values.

This is a thoughtful and definitely novel analysis that breaks new ground in that the panel of plasma tau biomarkers showed promise for serving as a proxy for brain imaging features of AD.

Replication of the findings in a separate cohort is a great strength, although as the authors surely know, both Biofinder and Triad are research cohorts that are likely to be relatively free of other comorbidities. But that is a limitation, not a weakness. These findings represent a proof of concept not a paradigm that is ready to be rolled out for broad clinical use.

This reviewer does not have issues with the methodology or the basic findings.

The authors may wish to clarify how their approach could be adopted by other places in that the classification scheme is based on values within the local cohort. What are the additional steps needed to take this concept beyond Biofinder and Triad?

The authors appropriately acknowledge that these analyses were conducted in persons in CU+, MCI+ and dementia+ who clinically, at least, had "pure AD pathology," and that a more heterogeneous sample would likely have attenuated the strength of associations because of the noise from co-morbidities.

It is obvious that the work presented here seeks to replace the research-only use of tau PET (or FDG PET) in tiny groups of patients as a means of depicting the topographic spread of AD neurodegeneration and extending it to anyone who walks into a clinic with a memory complaint. As a perspective that is more refined than one originating from, say, a ptau217 alone, this panel of plasma biomarkers offers the opportunity to differentiate between an early stage and a more advanced stage of brain tau pathology. Still this reviewer worries that plasma biomarker because of their ease of use carry the potential to be misused in clinical practice. The main threat comes from interpreting the plasma tau results without a thoughtfully approached clinical diagnosis. This is especially true for the distinction between CU and early MCI. Thus, I would like to see a more explicit statement by the authors that plasma tau biomarkers be interpreted only after a skilled examiner has established a clinical diagnosis of CU, MCI or dementia.

The limitations of a staging approach using any biomarker (including tau PET) is illustrated in Figure 3a – the results in comparison to syndromic diagnoses look terrific on a groupwise basis (highly significant chi sq) but its readily seen that neither the sensitivity, specificity, or PPV would be ready for clinical use. If the clinical differentiation between CU+ and MCI+ were not accurately established, a patient might erroneously be labeled as having an advanced symptomatic condition. And, by the way, may I implore the authors to avoid using the term "AD" by itself to signify dementia? Rather, please say amyloid positive dementia or Addem consistently, just as you do for MCI. Here, of all contexts, it is essential to keep syndrome (CU, MCI, dementia) clearly distinguishable from etiology (ie AD biology).

Reviewer #2

(Remarks to the Author)

The article, titled "PLASMA TAU BIOMARKERS FOR BIOLOGICAL STAGING OF ALZHEIMER'S DISEASE" carried out by Laia Montoliu-Gaya et al., is interesting. Exploration of related scientific issues is important to the field of Alzheimer's disease. However, it still falls short of meeting the criteria for acceptance based on the following concerns:

The statistical analysis method in this study rarely considers the impact of confounding factors such as age, gender, education, and APOE, both in the comparison of differences between diagnostic groups and in the correlation analysis.

These confounding factors are generally adjusted for as covariates in AD research. I would like to know whether the results

of this study remain robust after adjusting for these covariates.

In this study, AD dementia is defined clinically rather than by biomarkers, which is a significant limitation of this research. Table 1 should include a column for p-values to indicate the differences in basic characteristics between participants in the BioFinder-2 and TRIAD cohorts. Furthermore, the term "Non-AD impairment" needs clarification. Does it refer to Non-AD cognitive impairment or a Non-AD population? This needs to be explained in more detail.

The inclusion criteria for participants should be described in more detail, such as whether participants were recruited consecutively, the specific time period during which participants were recruited, and which patients were excluded. These details for both the BioFinder-2 and TRIAD cohorts need to be specified. Additionally, while it is mentioned that both studies were approved by their regional ethics committee, the specific ethics approval numbers should also be provided.

The abbreviation "Ab" should be corrected to "A β ". Please revise similar errors throughout the manuscript.

In the first paragraph of the discussion, the phrase "like their CSF counterparts" is not appropriate, as this study did not analyze CSF markers.

The lack of postmortem pathological validation is one of the limitations of this study.

Reviewer #3

(Remarks to the Author)

General Comments:

The establishment of novel biological staging systems represents a critical frontier in refining AD diagnosis and longitudinal pathological surveillance. While the authors present a multi-tau biomarker-derived stage model to delineate AD-related pathological dynamics, the clinical and mechanistic relevance of this framework remains insufficiently substantiated. Key limitations undermine its potential impact:

1. Unclear incremental value: The model's superiority over existing biomarker-driven classifications (e.g., NIA-AA AT(N) framework) is not conceptually articulated.

2. Underdeveloped clinical utility: Absence of ROC analyses, longitudinal predictive validations (e.g., progression to MCI/dementia), or comparisons with PET-based staging severely limits translational interpretation.

Specific Comments:

1. Inadequate stage discriminative power

While inter-stage biomarker differences reach statistical significance, the substantial overlap in biomarker distributions (e.g., interquartile ranges spanning adjacent stages) suggests limited categorical resolution.

2. Suboptimal data visualization

Figure 2 would benefit from:

o Overlaying LOESS curves ($\pm 95\%$ CI) for all CSF biomarkers to enable comparative trajectory analysis.

o Annotation of inflection points between stages to visualize putative pathological transitions.

3. Missing plasma biomarker dynamics

The study omits critical data on absolute plasma tau biomarker concentrations across stages. Providing:

o Quantitative cross-stage profiles (median, IQR) for plasma p-tau217r, p-tau205r, etc.

o Correlation matrices between plasma/CSF biomarkers at each stage would strengthen claims about peripheral biomarker utility.

4. Unvalidated predictive claims

The staging model requires validation against gold-standard endpoints:

o Pathological prediction: ROC analysis against amyloid-PET/tau-PET positivity (AUC metrics mandatory).

o Clinical progression: Cox regression models predicting time-to-cognitive decline with hazard ratios.

o Diagnostic specificity: Stage-wise differentiation of AD vs. non-AD tauopathies.

5. About p-tau217r definition

The operational definition of "p-tau217r" as the ratio of phosphorylated tau217 to non-p-212-221 requires rigorous clarification. Specifically:

o Does this differ from the pTau217/T-Tau217 ratio in PMID: 38514824 in epitope specificity (conceptual divergence) or assay methodology?

Version 1:

Reviewer comments:

Reviewer #1

(Remarks to the Author)

The authors have been responsive to my concerns and appeared to address adequately the concerns of other reviewers

Reviewer #2

(Remarks to the Author)

Thank you for your revision. Many of the concerns raised in the previous round have been adequately addressed, and I appreciate the authors' efforts in expanding the analyses and revising the manuscript. However, several important issues remain that should be clarified or improved:

(1) Thank you for including the additional analyses in the revised version. However, the corresponding results are currently

described in a rather brief or indirect manner, often referring readers to Extended Data Figures (e.g., Extended Data Figure 3 and 4) or Supplementary Tables without sufficient explanation in the main text. To enhance clarity and accessibility, I suggest that the key findings from these additional analyses be described in more detail in the main manuscript. This will help readers better understand the implications of the results without having to constantly refer to supplementary materials.

(2) I noticed that the updated analysis has adjusted for age, sex, and education; however, APOE status was not included as a covariate. In Alzheimer's disease research, APOE ϵ 4 carrier status is commonly considered an important confounding factor and is routinely adjusted for in both group comparisons and correlation analyses. Could you please clarify whether APOE genotyping was conducted in your study? If available, I strongly recommend including APOE status as a covariate in the analyses to ensure the robustness of the results.

(3) Additionally, the results after adjusting for all relevant covariates—including age, sex, education, and APOE—should be presented in the main figures, and the findings should be clearly described in the main text of the manuscript.

Reviewer #3

(Remarks to the Author)

After thoroughly reviewing the authors' response, I acknowledge that they have generally addressed the issues raised in my initial review. However, the following points still require further clarification:

1. Longitudinal predictive limitations: The inability to perform time-to-cognitive decline analyses due to retrospective plasma sampling is a significant limitation. While acknowledged, this issue undermines claims about the model's prognostic utility. We recommend emphasizing this caveat in the abstract and discussion, as it highlights the preliminary nature of the staging system's predictive capacity.

2. Correlation analysis and biomarker dynamics: The omission of correlation matrices between plasma biomarkers (e.g., p-tau217r vs. p-tau205r) within each stage weakens mechanistic interpretations. Even with small sample sizes per stage, exploratory correlations could reveal whether biomarker interactions differ across disease phases. A supplementary heatmap with stage-stratified correlations would strengthen claims about biomarker synergies.

3. PNS-Tau specificity: While PNS-tau was excluded from the staging model due to its lack of association with AD progression, the manuscript does not address whether peripheral comorbidities (e.g., renal dysfunction) influence its levels. A sensitivity analysis stratifying participants by comorbidities (e.g., eGFR) would clarify its utility as a negative control biomarker.

Version 2:

Reviewer comments:

Reviewer #2

(Remarks to the Author)

The authors have thoroughly addressed all of my comments, and the revised manuscript clearly indicates the changes with specific locations marked.

Reviewer #3

(Remarks to the Author)

I have carefully reviewed the authors' revisions addressing my previous comments and their responses. They have adequately addressed all of the issues and concerns I raised. The revisions to the manuscript are sufficient and effective, clarifying ambiguities I had previously noted and providing necessary supplementary information and arguments. I am satisfied with their detailed responses and the revised manuscript.

We very much thank the reviewers for their time and effort in reviewing our manuscript. Based on the insightful comments and suggestions of the reviewers, we have revised the manuscript and herein address each of their issues point by point. We believe these changes have significantly strengthened the manuscript. We look forward to the opportunity to communicate this data to the community.

Reviewer #1 (Remarks to the Author):

This is a two site discovery and replication exercise of a panel of plasma tau markers in an effort to develop a staging system in the plasma that proxies brain tau abundance and topographic spread. Using ptau217ratio to nonphosphorylated forms, ptau205ratio to nonphosphorylated forms and 0Ntau that were then normalized to their CU group, the authors performed a k-means clustering to create 3 groups. The authors did some further analyses to confirm relative stability of the classification. The key measure the biomarker stage derived from the 3 plasma tau biomarker values.

1. This is a thoughtful and definitely novel analysis that breaks new ground in that the panel of plasma tau biomarkers showed promise for serving as a proxy for brain imaging features of AD. Replication of the findings in a separate cohort is a great strength, although as the authors surely know, both Biofinder and Triad are research cohorts that are likely to be relatively free of other comorbidities. But that is a limitation, not a weakness. These findings represent a proof of concept not a paradigm that is ready to be rolled out for broad clinical use. This reviewer does not have issues with the methodology or the basic findings.

We sincerely appreciate the reviewer's thoughtful comment and their recognition that the absence of an analysis on the influence of comorbidities is, at this stage, a limitation rather than a weakness. We would like to clarify that BioFINDER is already a heterogeneous "real-world" cohort. Cognitively unimpaired (CU) individuals were recruited consecutively from a population-based study in Malmö, Sweden (PMID: 36224029). Additionally, patients with subjective cognitive decline (SCD), mild cognitive impairment (MCI), and Alzheimer's disease (AD) dementia were recruited from the memory clinics at Skåne University Hospital and Ängelholm Hospital in Sweden. These secondary care clinics receive referrals primarily from primary care (>80%), and the largest catchment area, Malmö, is a highly diverse city where approximately one-third of the population is foreign-born. Participants were enrolled as they presented to the clinics, representing a large range of educational backgrounds and comorbidities.

That said, we acknowledge the importance of evaluating the staging method in more cohorts and further considering the effect of comorbidities. We have now explicitly addressed this in the limitations section:

"Moreover, research involving more diverse cohorts, particularly those with a focus on peripheral comorbidities, is needed to evaluate how these comorbidities influence the model."

2. The authors may wish to clarify how their approach could be adopted by other places in that the classification scheme is based on values within the local cohort. What are the additional steps needed to take this concept beyond Biofinder and Triad?

We sincerely appreciate the reviewer's suggestion and the opportunity to elaborate on this point in the manuscript. To advance this concept beyond the BioFINDER and TRIAD cohorts, it will be important to evaluate the robustness of the model in more diverse populations with a wider range of comorbidities, as noted in the previous point. In addition, standardization of

methods and cut-offs, as well as assessing whether the approach remains valid when applied using other technologies such as immunoassays, will be crucial. We have added the following sentences to the limitations section of the manuscript:

“Finally, standardization of the staging model across various settings—through the establishment of cut-offs, common reference materials, and validation using different technologies—will be essential to advance this model toward clinical implementation.”

3. The authors appropriately acknowledge that these analyses were conducted in persons in CU+, MCI+ and dementia+ who clinically, at least, had “pure AD pathology,” and that a more heterogeneous sample would likely have attenuated the strength of associations because of the noise from co-morbidities.

We thank the reviewer for the positive comment.

4. It is obvious that the work presented here seeks to replace the research-only use of tau PET (or FDG PET) in tiny groups of patients as a means of depicting the topographic spread of AD neurodegeneration and extending it to anyone who walks into a clinic with a memory complaint. As a perspective that is more refined than one originating from, say, a ptau217 alone, this panel of plasma biomarkers offers the opportunity to differentiate between an early stage and a more advanced stage of brain tau pathology. Still this reviewer worries that plasma biomarker because of their ease of use carry the potential to be misused in clinical practice. The main threat comes from interpreting the plasma tau results without a thoughtfully approached clinical diagnosis. This is especially true for the distinction between CU and early MCI. Thus, I would like to see a more explicit statement by the authors that plasma tau biomarkers be interpreted only after a skilled examiner has established a clinical diagnosis of CU, MCI or dementia.

We appreciate the reviewer's insight and the reminder of the importance of clearly stating the role of a skilled examiner in clinical diagnosis. We agree that the proposed staging framework should serve as a complementary tool to a physician's evaluation, never as a stand-alone method. The following sentence has been added to the discussion:

“This staging model could serve as a valuable first-line screening tool that, when assessed by a skilled examiner, may complement clinical cognitive diagnosis and improve patient management in clinical settings.”

5. The limitations of a staging approach using any biomarker (including tau PET) is illustrated in Figure 3a – the results in comparison to syndromic diagnoses look terrific on a groupwise basis (highly significant chi sq) but its readily seen that neither the sensitivity, specificity, or PPV would be ready for clinical use. If the clinical differentiation between CU+ and MCI+ were not accurately established, a patient might erroneously be labeled as having an advanced symptomatic condition.

We appreciate the reviewer's opportunity to clarify this point. Our aim with this paper is not to provide a staging model that is ready for clinical implementation. Rather, we present it as a proof-of-concept demonstrating that tau blood biomarkers can be used for AD staging. However, we acknowledge that we do not foresee a direct clinical application of this staging model on its own, but rather as a tool to aid physicians in patient management.

To clarify this further, we have added in the Discussion section the sentence specified in the previous point:

“This staging model could serve as a valuable first-line screening tool that, when assessed by a skilled examiner, may complement clinical cognitive diagnosis and improve patient management in clinical settings.”

And the following in the limitations section:

“In this regard, while the staging system may be effective at the group level, further work is needed to improve its specificity for potential clinical use.”

6. And, by the way, may I implore the authors to avoid using the term “AD” by itself to signify dementia? Rather, please say amyloid positive dementia or Addem consistently, just as you do for MCI. Here, of all contexts, it is essential to keep syndrome (CU, MCI, dementia) clearly distinguishable from etiology (ie AD biology).

We thank the reviewer for bringing to our attention this valuable point. We have changed AD to ADdem throughout the manuscript.

Reviewer #2 (Remarks to the Author):

The article, titled "PLASMA TAU BIOMARKERS FOR BIOLOGICAL STAGING OF ALZHEIMER'S DISEASE" carried out by Laia Montoliu-Gaya et al., is interesting. Exploration of related scientific issues is important to the field of Alzheimer's disease. However, it still falls short of meeting the criteria for acceptance based on the following concerns:

1. The statistical analysis method in this study rarely considers the impact of confounding factors such as age, gender, education, and APOE, both in the comparison of differences between diagnostic groups and in the correlation analysis. These confounding factors are generally adjusted for as covariates in AD research. I would like to know whether the results of this study remain robust after adjusting for these covariates.

We thank the reviewer for bringing this to our attention. We have now included key covariates—age and sex (and years of education for cognitive outcomes)—in our analyses. The updated results are presented in Supplementary Tables 8, 9 and 10 (see below). We confirm that the observed differences remain robust and are not driven by these covariates. The description of these extra analysis has been added to the method section:

“A sensitivity analysis using ANCOVA adjusted by age and sex is included in Supplementary Table 8”.

“A sensitivity analysis using ANCOVA adjusted by age and sex and years of education for cognitive outcomes) for continuous variables is included in Supplementary Table 9”.

“In a sensitivity analysis, age and sex (and years of education for cognitive outcomes) were used as covariates (Supplementary Table 10).”

Supplementary Table 8. Differences in plasma biomarkers by diagnosis within the Alzheimer's Disease (AD) continuum.

	CU- vs. CU+		CU+ vs. MCI+		MCI+ vs. ADdem+	
	β -coefficient	p-value	β -coefficient	p-value	β -coefficient	p-value
p_181_r	-0.40	0.007	-0.20	0.496	-0.35	0.067
p_205_r	-0.52	<0.001	-0.56	<0.001	-0.50	<0.001
p_217_r	-0.78	<0.001	-0.54	<0.001	-0.71	<0.001
p_231_r	-0.80	<0.001	-0.30	0.059	-0.47	0.001
p_199_r	0.06	0.975	0	1	-0.06	0.979
p_202_r	0.24	0.291	0.11	0.905	0.01	1
p-tau181	-0.48	<0.001	-0.27	0.158	-0.60	<0.001
Tau 212-221	0.04	0.994	-0.20	0.581	-0.15	0.765
p-tau217	-0.59	<0.001	-0.56	<0.001	-0.78	<0.001
Tau 195-209	-0.18	0.472	-0.28	0.248	-0.32	0.146
p-tau199	-0.15	0.689	-0.10	0.912	-0.28	0.268
p-tau202	-0.03	0.996	-0.11	0.897	-0.24	0.435
p-tau205	-0.43	<0.001	-0.54	<0.001	-0.6	<0.001
p-tau231	-0.67	<0.001	-0.39	0.005	-0.63	<0.001
Tau-0N	-0.40	0.005	-0.23	0.355	-0.54	0.001
Tau-1N	-0.13	0.746	-0.12	0.862	-0.38	0.062
PNS-tau 131	0.17	0.599	-0.18	0.669	-0.14	0.825
PNS-tau275	0.04	0.994	-0.07	0.969	-0.08	0.966

Supplementary Table 9. Differences in AD biomarkers by plasma stages in the BioFINDER-2 and TRIAD cohorts.

	Stage 1 vs. 0		Stage 2 vs. 1		Stage 3 vs. 2	
	β -coefficient	p-value	β -coefficient	p-value	β -coefficient	p-value
BioFINDER-2						
Aβ-PET	-0.45	<0.001	-1.40	<0.001	0.02	1
Tau-PET early region (MTL)	-0.21	0.060	-1.36	<0.001	-0.29	0.153
Tau-PET intermediate region (NeoT)	-0.12	0.503	-1.15	<0.001	-0.99	<0.001
Cortical thickness	0.25	0.048	0.82	<0.001	0.32	0.164
mPACC	0.15	0.370	0.90	<0.001	0.58	0.004
MMSE	0.10	0.658	1.01	<0.001	0.59	<0.001
TRIAD						
Aβ-PET	-1.45	<0.001	-0.53	0.001	0.70	0.157
Tau-PET early region (MTL)	-0.80	<0.001	-0.90	<0.001	-0.21	0.969
Tau-PET intermediate region (NeoT)	-0.50	0.005	-1.11	<0.001	-1.14	0.045
Cortical thickness	0.55	0.007	0.98	<0.001	0.12	0.995
CDR-SB	-0.67	0.001	-0.62	0.016	-0.45	0.814
MMSE	0.53	0.013	0.57	0.036	1.29	0.069

Supplementary Table 10. Differences in slope of AD biomarkers by plasma stages.

	Stage 1 vs. 0		Stage 2 vs. 1		Stage 3 vs. 2	
	β -coefficient	p-value	β -coefficient	p-value	β -coefficient	p-value
Aβ-PET	-0.57	<0.001	-0.33	0.166	-0.01	1
Tau-PET early region (MTL)	-0.17	0.178	-1.35	<0.001	-0.24	0.311
Tau-PET intermediate region (NeoT)	-0.09	0.702	-1.09	<0.001	-0.99	<0.001
Cortical thickness	0.23	0.178	0.83	<0.001	0.61	0.001
mPACC	0.08	0.847	1.00	<0.001	0.46	0.005
MMSE	0.06	0.933	0.85	<0.001	0.63	<0.001

2. In this study, AD dementia is defined clinically rather than by biomarkers, which is a significant limitation of this research.

We apologize for not making this clear in the submitted manuscript. MCI and dementia patients were required to be A β -positive, as determined by either CSF or PET, to be considered AD. We have revised the Methods section to explicitly include this criterion for both cohorts, BioFINDER and TRIAD:

“MCI and dementia patients had to be A β -positive by either CSF or PET to be considered due to AD.”

3. Table 1 should include a column for p-values to indicate the differences in basic characteristics between participants in the BioFinder-2 and TRIAD cohorts.

We thank the reviewer for this suggestion. We have now included a column in Table 1 for the p-values comparing the basic characteristics between BioFinder-2 and TRIAD. See below:

	BioFINDER-2 (n=549)	TRIAD (n=140)	p-value
Age, years	70.4 (12.5)	71.5 (6.5)	0.140
Women, n(%)	284 (51.7%)	58 (41.4%)	0.176
Education years	12.4 (3.7) [n=545]	15.6 (3.49)	<0.001
APOE ϵ 4 carriership, n(%)	281 (51.3%) [n=548]	61 (43.6%)	0.125
Diagnosis, n(%)			<0.001
CU-	191 (34.8%)	34 (24.3%)	
CU+	82 (14.9%)	25 (17.9%)	
MCI+	76 (13.8%)	35 (25.0%)	
AD dementia	80 (14.6%)	28 (20.0%)	
Non-AD impairment	113 (20.6%)	7 (5.0%)	
Other	7 (1.3%)	11 (7.9%)	

A β -positivity, n(%)	256 (47.2%) [n=542]	88 (62.9%) [n=129]	<0.001
Tau-PET positivity, n(%)	150 (27.5%) [n=545]	54 (38.6%) [n=127]	0.001

4. Furthermore, the term "Non-AD impairment" needs clarification. Does it refer to Non-AD cognitive impairment or a Non-AD population? This needs to be explained in more detail.

We thank the reviewer for the opportunity to clarify this concept. 'Non-AD impairment' refers to patients with cognitive impairment not attributed to AD, based on clinical diagnosis and biomarker status. Throughout the text, we have revised this terminology to 'Cognitively Impaired not due to AD.' Additionally, we have further clarified this in the Methods section, specifying:

"Participants with other etiological diagnosis than AD were classified as non-AD and are referred as Cognitively Impaired not due to AD".

5. The inclusion criteria for participants should be described in more detail, such as whether participants were recruited consecutively, the specific time period during which participants were recruited, and which patients were excluded. These details for both the BioFinder-2 and TRIAD cohorts need to be specified. Additionally, while it is mentioned that both studies were approved by their regional ethics committee, the specific ethics approval numbers should also be provided.

The following information has been added to the Method section for each cohort:

BioFinder: All participants provided written informed consent and were recruited consecutively without excluding any eligible participants between March 2019 and November 2022.(...) The study was approved by the Regional Ethics Committee in Lund, Sweden (Dnr 2016-1053).

TRIAD: TRIAD patients were approached consecutively, and no eligible participants were excluded. Data from the TRIAD study was collected between October 2019 and May 2024 (...) The study was approved by the Montreal Neurological Institute PET working committee and the Douglas Mental Health University Institute Research Ethics Board (Ethical approval: MP-18-2019-223, IUSMD-19-05).

6. The abbreviation "Ab" should be corrected to "A β ". Please revise similar errors throughout the manuscript.

We thank the reviewer for detecting this mistake. We have exchanged Ab to A β throughout the text.

7. In the first paragraph of the discussion, the phrase "like their CSF counterparts" is not appropriate, as this study did not analyze CSF markers.

We thank the reviewer for this comment. When we referred to CSF biomarkers, it was not in relation to the results in this manuscript, but in reference to our previous publications (PMID: 38514824). To avoid confusion, we have deleted "like their CSF counterparts" from the sentence. Please find the new version below:

Our results demonstrate that different plasma tau biomarkers can be used to stage AD biologically in a clinically relevant manner, as we previously showed in CSF.

8. The lack of postmortem pathological validation is one of the limitations of this study. We agree with the reviewer that the lack of postmortem pathological validation is a limitation of the study. To acknowledge this, we have added it to the limitations section:

Future studies should validate the staging model in cohorts with postmortem neuropathological examination (...).

Reviewer #3 (Remarks to the Author):

General Comments:

The establishment of novel biological staging systems represents a critical frontier in refining AD diagnosis and longitudinal pathological surveillance. While the authors present a multi-tau biomarker-derived stage model to delineate AD-related pathological dynamics, the clinical and mechanistic relevance of this framework remains insufficiently substantiated. Key limitations undermine its potential impact:

1. Unclear incremental value: The model's superiority over existing biomarker-driven classifications (e.g., NIA-AA AT(N) framework) is not conceptually articulated.

We appreciate the opportunity to elaborate further on this point. The ATN framework, published in 2018 (PMID: 29653606), serves as a diagnostic tool aimed at categorizing AD based on three core biomarkers: amyloid, tau, and neurodegeneration (ATN). Recently, the Alzheimer's Association revised this framework to address new developments in AD biomarker research. The updated criteria (PMID: 38934362), not only refine the guidelines for AD diagnosis, incorporating a broader range of biomarkers, but it also introduces a novel framework for staging AD based on biological biomarkers—a feature that was absent in the original ATN framework. Unlike the ATN framework, which is designed primarily as a diagnostic tool for identifying AD, the new staging framework is intended to track the progression of the disease. While both frameworks rely on biomarkers, their applications differ. The staging framework helps determine the stage of AD in a patient, providing crucial insights into the progression of the disease, while the ATN framework is focused on diagnosis.

The staging framework described in the new guidelines is divided into Imaging and Fluid biomarkers – being the framework for imaging and fluid biomarkers not equivalent. In the imaging criteria individuals are categorized based on their negative or positive A β -PET status, and tau-PET uptake in MTL and NeoT regions as follows: negative for both amyloid and tau-PET (A-T2-), initial (A β -PET-positive and Tau-PET-negative, A+T2), early (A β -PET-positive and Tau-PET MTL-positive, A+T2MTL+), intermediate (A β -PET-positive and Tau-PET MTL- and NeoT-positive, A+T2MOD+), and advanced (A β -PET-positive and Tau-PET MTL- and NeoT-positive with high uptake in NeoT, A+T2MOD+A+T2HIGH+). The proposed staging system for fluid biomarkers is described as conceptual and considered to need further validation.

In recent years, significant advancements in the study of fluid tau variants have emerged, as discussed in the introduction. This has spurred interest in utilizing tau fluid biomarkers for staging AD. In this paper we aim to evaluate if plasma tau biomarkers are useful to stage the disease, as already proven with CSF biomarkers and as proposed by the new AA guidelines. It is important to note, that although CSF biomarkers have shown high accuracy on this regard, being able to stage AD using plasma biomarkers will significantly decrease patients' burden and economic costs. To make this point clearer we have added the following sentences to the introduction:

“The Alzheimer's Association (AA) recently published revised criteria for the diagnosis of AD, updating the previous guidelines published in 2018. These new criteria also include a

biomarker framework for staging AD using amyloid and tau-PET imaging biomarkers, as well as a conceptual biological staging model based on fluid biomarkers. However, further investigation and validation are necessary to confirm the utility of fluid tau biomarkers, particularly those in blood, for staging AD—an advancement that would be highly valuable for patient management.”

2. Underdeveloped clinical utility: Absence of ROC analyses, longitudinal predictive validations (e.g., progression to MCI/dementia), or comparisons with PET-based staging severely limits translational interpretation.

We thank the reviewer for the suggestion to perform extra analysis to consolidate our message. We have now included LOESS and ROC analysis (see points 2 and 4 in Specific Comments). For longitudinal predictive validations see point 4 in Specific Comments. Additionally, the manuscript already included comparisons with PET imaging (Figure 3 and Extended Data Figure 4).

Specific Comments:

1. Inadequate stage discriminative power

While inter-stage biomarker differences reach statistical significance, the substantial overlap in biomarker distributions (e.g., interquartile ranges spanning adjacent stages) suggests limited categorical resolution.

We thank the reviewer for the opportunity to elaborate on this point. While there is indeed some overlap in AD biomarkers across different plasma stages, our data demonstrate a clear trend toward increased abnormality in later plasma stages. More importantly, these abnormalities emerge at different stages for different biomarkers, which aligns with findings from previous literature. This pattern suggests that, although plasma stages do not align perfectly with imaging or other AD biomarkers—as already acknowledged in the revised criteria proposed by Jack et al. 2024 (PMID:38934362)—they are related. This relationship supports the potential clinical utility of plasma staging, as further evidenced by our ROC curve analyses (see response to Point #4).

2. Suboptimal data visualization

Figure 2 would benefit from:

- o Overlaying LOESS curves ($\pm 95\%$ CI) for all CSF biomarkers to enable comparative trajectory analysis.

We thank the reviewer for this suggestion. We have now plotted the locally estimated scatterplot smoothing (LOESS) curves showing changes in each plasma biomarker included in the model, alongside plasma stages and the corresponding changes in imaging and cognitive biomarkers at each plasma stage. Since Figure 2 was becoming too crowded, we have created a new figure (Extended Data Figure 3) to better illustrate these findings. Additionally, we have included the relevant details in the Results section:

“Locally estimated scatterplot smoothing (LOESS) representation of the change of the biomarkers included in the model together with the changes in imaging and cognitive

biomarkers with plasma stages can be visualized in Extended Data Figure 3.

Extended Figure 3. AD biomarker trajectories by plasma stages in the BioFINDER-2 cohort.

LOESS regression of AD biomarkers by plasma stages. The left figure shows imaging and cognitive biomarkers, the right figure shows plasma biomarkers included in the staging model plus plasma PNS-tau for comparison. Biomarker levels are z-scored based on a group of CU- participants, and all increases represent an increase in abnormality. Colored lines and bands represent the LOESS regression and its 95% CI. Horizontal line is drawn at z-score = 1.96, which represents 95% CI of the reference group (CU-). LOESS, locally estimated scatterplot smoothing. MMSE, Mini-Mental State Examination; mPACC, modified preclinical Alzheimer’s cognitive composite; MTL, medial temporal lobe; NeoT: temporal neocortex.

o Annotation of inflection points between stages to visualize putative pathological transitions.

In the aforementioned new Extended Figure 3 (see above), we have included a dotted line at z-score=1.96 (corresponding to 95%CI of the reference group), to show when these biomarkers become abnormal in comparison to reference group (defined as CU- individuals).

3. Missing plasma biomarker dynamics

The study omits critical data on absolute plasma tau biomarker concentrations across stages. Providing:

o Quantitative cross-stage profiles (median, IQR) for plasma p-tau217r, p-tau205r, etc.

We agree with the reviewer that this is critical information that was missing in the previous version of the manuscript. We have included this information in Supplementary Table 4:

Plasma stages	0	1	2	3	All
BioFINDER					
p-tau217r	0.52 [0.39, 0.69]	0.81 [0.67, 1.21]	2.72 [2.28, 3.19]	3.66 [2.89, 4.7]	0.77 [0.52, 2.08]
p-tau205r	0.014 [0.011, 0.017]	0.021 [0.017, 0.025]	0.029 [0.025, 0.033]	0.035 [0.029, 0.04]	0.018 [0.014, 0.026]
Tau-ON	1.99 [1.46, 2.65]	3.98 [3.31, 4.93]	3.66 [2.68, 4.58]	7.60 [6.08, 8.95]	2.86 [1.96, 4.32]
TRIAD					

p-tau217r	0.71 [0.48, 0.9]	1.88 [1.47, 2.12]	3.12 [2.87, 3.81]	5.06 [4.96, 5.25]	1.36 [0.77, 2.42]
p-tau205r	0.004 [0.003, 0.005]	0.007 [0.006, 0.009]	0.01 [0.009, 0.013]	0.017 [0.013, 0.017]	0.006 [0.004, 0.009]
Tau-0N	3.56 [2.57, 5.01]	5.79 [4.29, 8.17]	8.7 [7.06, 12.62]	14.85 [13.83, 16.11]	5.21 [3.31, 8.4]

o Correlation matrices between plasma/CSF biomarkers at each stage would strengthen claims about peripheral biomarker utility.

We did not have CSF biomarker data available for all participants - A β positivity was defined based on either CSF or PET biomarkers (see Methods section). Given that our panel includes both CNS- and PNS-specific tau peptides, we conducted correlation analyses among all biomarkers measured in the panel (see below). Our results revealed a strong correlation among previously established plasma p-tau biomarkers, such as p-tau181, p-tau231, and p-tau217. In contrast, PNS-tau exhibited low correlations with all other biomarkers, but the targeted PNS-specific peptides showed a high correlation between them. Similarly, p-tau199 and p-tau202 presented generally low correlations with other biomarkers, though they were significantly correlated with each other. Interestingly, the CNS-specific peptides also showed high correlations between them. While these findings are intriguing, we acknowledge that their interpretation remains limited at this stage and consider them inconclusive.

We hope our response adequately addresses the reviewer's concern. However, we are unsure about the specific rationale behind analyzing the data by disease stage. In any case, the number of participants per stage is relatively small, which would substantially reduce the statistical power and limit the interpretability and robustness of the correlation results.

4. Unvalidated predictive claims

The staging model requires validation against gold-standard endpoints:

o Pathological prediction: ROC analysis against amyloid-PET/tau-PET positivity (AUC metrics mandatory).

We appreciate the reviewer's comment regarding the validation of our staging model against gold standard endpoints. While we acknowledge the importance of such comparisons, it is important to note that plasma biomarkers are not expected to capture the exact same pathological processes as CSF or PET biomarkers — a distinction that has been explicitly recognized in the updated AA research framework for AD (PMID: 38934362). Nonetheless, to address this point, we have included these comparisons in the revised version of the manuscript (Extended Data Figure 4). Specifically, for A β -status (assessed via CSF A β 42/40 ratio or PET imaging, noting that A β -PET was not available for most dementia patients), our model achieved an AUC [95% CI] of 0.84 [0.81–0.87]. For Tau-PET, the AUC [95% CI] was 0.90 [0.87–0.93].

Extended Figure 4. Plasma stages for predicting dichotomic categories. ROC curves were used to assess the classification of A β -PET, tau-PET and AD versus non-AD cognitive impairment. AUC, area under the curve; ROC, receiver operating characteristic.

o Clinical progression: Cox regression models predicting time-to-cognitive decline with hazard ratios.

We agree with the reviewer that it would be preferable to assess the prognostic value of our staging model against disease progression. Unfortunately, the availability of prospective longitudinal data is restricted as most of the plasma measures were obtained at the second visit of our cohort, and not at baseline. Therefore, the sample size is not big enough to perform such analyses. We have added a mention to this limitation in our discussion:

Furthermore, time-to-cognitive decline analysis could not be performed due to the limited availability of prospective longitudinal data as most of the plasma measures were obtained at the second visit of the BioFINDER-2 cohort, and not at baseline.

o Diagnostic specificity: Stage-wise differentiation of AD vs. non-AD tauopathies.

We have included in the manuscript the differentiation of AD vs non-AD etiology in individuals with cognitive symptoms (MCI or dementia stages; Extended Data Figure 4): AUC [95%CI]= 0.88 [0.84-0.92] (Extended data Figure 4).

Extended Figure 4. Plasma stages for predicting dichotomic categories. ROC curves were used to assess the classification of Aβ-PET, tau-PET and AD versus non-AD cognitive impairment. AUC, area under the curve; ROC, receiver operating characteristic.

5. About p-tau217r definition

The operational definition of "p-tau217r" as the ratio of phosphorylated tau217 to non-p-212-221 requires rigorous clarification. Specifically:

- o Does this differ from the pTau217/T-Tau217 ratio in PMID: 38514824 in epitope specificity (conceptual divergence) or assay methodology?

The p-tau217/non-p-tau217 ratio used in this work (here referred to p-tau217r) does not differ from the one used in the previous publication (PMID: 38514824), so the specificity of the epitope remains the same. The sequence and other specific details of the peptides targeted in this study can be found in Supplementary Table 7.

Supplementary Table 7. Tryptic tau peptides targeted in the study.

Respective peptide sequence, dominant charge state and monoisotopic m/z value.

Peptide	Peptide aa positions	Target peptide sequence	Charge state	m/z
P-tau181	175-190	TPPAPK[pT]PPSSGEPK	3	556.606
Tau195-209	195-209	SGYSSPGSPGTPGSR	2	697.321
P-tau199	195-209	SGYS[pS]PGSPGTPGSR	2	737.304
P-tau202	195-209	SGYSSPG[pS]PGTPGSR	2	737.304
P-tau205	195-209	SGYSSPGSPG[pT]PGSR	2	737.304
Tau212-221	212-221	TPSLPTPPTTR	2	533.798
P-tau217	212-221	TPSLP[pT]PPTTR	2	573.781
P-tau231	225-240	KVAVVR[pT]PPKSPSSAK	3	577.989
0N-tau	045-126	AEEAGIGDTPSLEDEAAGHVTQAR	3	808.714
1N-tau	068-126	STPTAEAEAEAGIGDTPSLEDEAAGHVTQAR	3	1004.134
PNS 131-138	131-138	VVQEGFLR	2	474.269
PNS 275-291	275-291	VSTEIPASEPDGPSVGR	2	849.421

The information on how the peptide ratios are calculated is included in the results section:

Apart from the quantified peptides, we additionally included in the analysis the ratios between the phosphorylated peptides p-tau217 and p-tau205 and their respective non-phosphorylated forms: p-tau217/212-221 (p-tau217r) and p-tau205/195-209 (p-tau205r).

Regarding methodological differences, the main one lies in the sample preparation, as the current study was performed in plasma, whereas the previous one was conducted in CSF. In the earlier study, tau was enriched through immunoprecipitation using Tau 1 and the HJ antibody series (HJ8.5, HJ8.7, HJ32.11, and HJ34.8). In contrast, the current study employed a combination of Tau12, HT7, and BT2 antibodies for immunoprecipitation, as previously described (PMID: 37198279). We believe the MS parameters were largely similar in both studies. However, we do not have access to all MS parameters in the previous publication, as this information has not been published.

We thank the reviewers for their thoughtful input during this second round of review, which has helped us further improve our manuscript. In response to the insightful comments and suggestions provided, we have revised the manuscript accordingly and addressed each point below in detail. We believe these changes have significantly strengthened the work and we look forward to the opportunity to communicate this data to the community.

Reviewer #2 (Remarks to the Author):

Thank you for your revision. Many of the concerns raised in the previous round have been adequately addressed, and I appreciate the authors' efforts in expanding the analyses and revising the manuscript. However, several important issues remain that should be clarified or improved:

(1) Thank you for including the additional analyses in the revised version. However, the corresponding results are currently described in a rather brief or indirect manner, often referring readers to Extended Data Figures (e.g., Extended Data Figure 3 and 4) or Supplementary Tables without sufficient explanation in the main text. To enhance clarity and accessibility, I suggest that the key findings from these additional analyses be described in more detail in the main manuscript. This will help readers better understand the implications of the results without having to constantly refer to supplementary materials.

We appreciate the reviewer's observation and agree that, in some instances, the Supplementary Material was not described in sufficient detail. We have made efforts to clarify and expand these descriptions where possible. That said, providing more extensive information is somewhat constrained by the article's word limit, but we have aimed to strike a better balance. Examples:

"Locally estimated scatterplot smoothing (LOESS) representation of the change of the plasma biomarkers at each stage showed that p-tau217r p-tau205r and 0N-tau become abnormal consecutively and in accordance with the plasma stages (Extended Data Fig. 3)"

"Regarding the plasma stages, A β -PET showed the earliest changes toward abnormality, followed by Tau-PET MTL, Tau-PET NeoT, cognitive scores, and finally, cortical thickness (Extended Data Figure 6). In addition, the model was proven accurate to predict A β -PET (AUC [95%CI] = 84 [81-87]) and tau-PET (AUC [95%CI] = 90 [87-93]) status as well as AD-related cognitive symptoms (AUC [95%CI] = 88 [84-92]) (Extended Data Fig. 7)."

"A consistent pattern of changes in AD pathology biomarkers was observed across stages in both the BioFINDER-2 and TRIAD cohorts, supporting the validity of the model (representation in Figure 2, quantitative cross-stage profiles in Supplementary Table 4 and analysis of group differences in Supplementary Table 5).

(2) I noticed that the updated analysis has adjusted for age, sex, and education; however, APOE status was not included as a covariate. In Alzheimer's disease research, APOE ϵ 4 carrier status is commonly considered an important confounding factor and is routinely adjusted for in both group comparisons and correlation analyses. Could you please clarify whether APOE genotyping was conducted in your study? If available, I strongly recommend including APOE status as a covariate in the analyses to ensure the robustness of the results.

We thank the reviewer for this valuable suggestion. APOE status has now been included as a covariate in the main analysis. The Results, Figures, and Tables have been updated accordingly. Corresponding changes have also been made to the Methods section and figure legends.

(3) Additionally, the results after adjusting for all relevant covariates—including age, sex, education, and APOE—should be presented in the main figures, and the findings should be clearly described in the main text of the manuscript.

The adjusted results have been incorporated into the main text and primary figures, with all numerical details provided in the supplementary tables.

Reviewer #3 (Remarks to the Author):

After thoroughly reviewing the authors' response, I acknowledge that they have generally addressed the issues raised in my initial review. However, the following points still require further clarification:

1. Longitudinal predictive limitations: The inability to perform time-to-cognitive decline analyses due to retrospective plasma sampling is a significant limitation. While acknowledged, this issue undermines claims about the model's prognostic utility. We recommend emphasizing this caveat in the abstract and discussion, as it highlights the preliminary nature of the staging system's predictive capacity.

We thank the reviewer for this valuable suggestion.

As noted, this point was addressed in the Limitations section, which we have now expanded further.

“Additionally, since the blood data was not acquired at baseline but at the second visit of the BioFINDER-2 cohort, the available follow-up data was insufficient to perform purely prospective analyses. Therefore, we also included retrospective data in our longitudinal analyses. In this regard, time-to-cognitive decline analysis could not be performed due to the limited availability of prospective longitudinal data. We acknowledge the potential limitations and biases introduced by this approach, and further research is needed to validate these findings. Furthermore, the results were evaluated at the group level, limiting direct clinical application. In this regard, while the staging system may be effective at the group level, further work is needed to improve its specificity for potential clinical use.”

We have also made changes to the Discussion to better reflect the limitations of our findings. For example, we replaced the word "validate" with "support" in the following sentence:

"Overall, our findings support the utility of distinct blood tau biomarkers for staging AD, offering significant potential for use in both clinical practice and trials."

We also revised the following paragraph in the Discussion to improve clarity and nuance:

“Additionally, each plasma stage was associated with a different mean rate of cognitive decline. This capability could assist clinicians to forecast individual deterioration based on plasma stage. For example, the efficacy of anti-amyloid therapies to delay cognitive decline has been suggested to be affected not only by tau-PET levels, but also cognitive status. Individuals with same low to medium tau-PET burden, but more advanced clinical dementia ratings may have fewer additional months of independence with the same treatment. However, we acknowledge that our analyses should be replicated in future studies using exclusively prospective data and assess how accurately plasma stages can predict time-to-decline. This could significantly enhance clinical decision-making and patient care.”

Due to space limitations, we were unable to include this in the abstract.

2. Correlation analysis and biomarker dynamics: The omission of correlation matrices

between plasma biomarkers (e.g., p-tau217r vs. p-tau205r) within each stage weakens mechanistic interpretations. Even with small sample sizes per stage, exploratory correlations could reveal whether biomarker interactions differ across disease phases. A supplementary heatmap with stage-stratified correlations would strengthen claims about biomarker synergies.

We have now included the correlation matrices for all individuals and for individuals at each plasma stage in Extended Data Figure 4.

3. PNS-Tau specificity: While PNS-tau was excluded from the staging model due to its lack of association with AD progression, the manuscript does not address whether peripheral comorbidities (e.g., renal dysfunction) influence its levels. A sensitivity analysis stratifying participants by comorbidities (e.g., eGFR) would clarify its utility as a negative control biomarker.

We thank the reviewer for suggesting this analysis, which has significantly improved the manuscript. As suggested, we performed a sensitivity analysis by stratifying participants according to eGFR. The levels of all tau biomarkers, including PNS-tau, were significantly

altered in individuals with kidney dysfunction. However, this effect was mitigated when using phosphorylated/non-phosphorylated tau ratios. We believe that chronic kidney disease leads to an overall increase in blood protein levels, including peripheral tau, which limits the utility of PNS-tau as a negative control biomarker. Nevertheless, this confounding effect appears to be reduced when using tau ratios.

We have incorporated the analysis in Supplementary Table 2:

Biomarker	Estimate_ CKDneg - CKDpos	p-value_ CKDneg - CKDpos
p-tau205_r	-0.09	0.377
p-tau217_r	0.04	0.688
p-tau181	-0.62	<0.001
Tau 212-221	-0.60	<0.001
Tau 195-209	-0.67	<0.001
p-tau199	-0.59	<0.001
p-tau202	-0.46	<0.001
p-tau231	-0.48	<0.001
Tau-0N	-0.31	0.003
Tau-1N	-0.48	<0.001
PNS-tau 131-138	-0.36	0.001
PNS-tau 275-291	-0.34	0.002

The following has been included in the results section:

In addition, the ratios p-tau217r and p-tau205r were selected over the phosphorylated tau species alone because ratios of p-tau to non-phosphorylated tau may help minimize confounding effects from medical comorbidities such as kidney disease. To explore this, we performed a sensitive analysis stratifying participants by Chronic Kidney Dysfunction (CKD) status based on eGFR measures (positivity: <60 mL/min/1.73m², Supplementary Table 2). We found that all tau biomarkers, including PNS-tau, were significantly altered in individuals with kidney dysfunction, but this effect was mitigated when using the phospho/non-phospho ratios, corroborating previous findings³³.

And in the Discussion:

For p-tau217 and p-tau205, we employed the ratios of phosphorylated to non-phosphorylated peptides (tau212-221 and tau195-209, respectively), as we show that these ratios can mitigate the impact of elevated tau levels in individuals with chronic kidney dysfunction, corroborating previous results^{33,40}, and had greater consistency across cohorts.